# NestQuant: Nested Lattice Quantization for Matrix Products and LLMs

**Semyon Savkin** [*1]   **Eitan Porat** [*2]   **Or Ordentlich** [3]   **Yury Polyanskiy** [1]

## Abstract

Post-training quantization (PTQ) has emerged as a critical technique for efficient deployment of large language models (LLMs). This work proposes NESTQUANT, a novel PTQ scheme for weights and activations that is based on self-similar nested lattices. Recent works have mathematically shown such quantizers to be information-theoretically optimal for low-precision matrix multiplication. We implement a practical low-complexity version of NestQuant based on Gosset lattice, making it a drop-in quantizer for any matrix multiplication step (e.g., in self-attention, MLP etc). For example, NestQuant quantizes weights, KV-cache, and activations of Llama-3-8B to 4 bits, achieving perplexity of 6.6 on wikitext2. This represents more than 55% reduction in perplexity gap with respect to unquantized model (perplexity of 6.14) compared to state-of-the-art Meta's SpinQuant (perplexity 7.3), OstQuant (7.3) and QuaRot (8.2). Comparisons on bigger models (up to 70B) and on various LLM evaluation benchmarks confirm uniform superiority of NestQuant.

## 1. Introduction

There are three principal goals of post-training quantization (PTQ). First, reducing the number of bits per parameter allows for loading big models on cheap GPUs with limited memory, thus democratizing access to LLMs. This requires "weights-only" quantization algorithms of which the most popular are AWQ, GPTQ, and QuIP (see references in Section 2.2).

The second goal of PTQ is to accelerate inference. In LLMs most of the compute is spent multiplying matrices. Multiply-

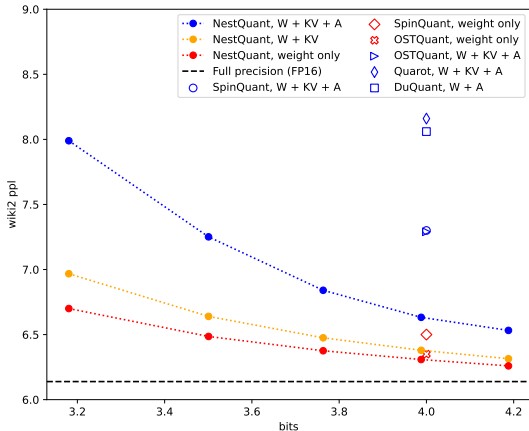

*Figure 1.* Perplexity of quantized Llama-3-8B models for three regimes (weight-only, weights + KV cache, end-to-end) on wikitext2 vs number of bits per entry.

ing a pair of $n \times n$ matrices requires $2n^3$ FLOPs and $\frac{3}{8}Rn^2$ bytes to exchange between the core and memory (here and below $R$ designates the number of bits required to store each entry of a vector/matrix). So when matrices are large (such as during the pre-fill phase when the prompt is processed) the GPU is compute-bound, while when $n$ is small (such as during generation) the GPU becomes memory-bound. One therefore needs to reduce $R$ by quantizing both weights and the KV cache, thereby alleviating the memory bandwidth bottleneck during generation.

The third goal of PTQ is to accelerate inference of giant LLMs that require hosting each layer on a separate GPU (pipelining parallelism). For this goal one needs to quantize activations passed from one layer to the next to reduce the communication bottleneck.

While quantization of weights to $R = 3, 4$ and even $R = 2$ bits has been achieved with minimal loss of quality, quantization of KV cache and activations has been much more challenging. Popular algorithms for full quantization are LLM.int8(), SmoothQuant and SpinQuant (see references in Section 2.2), the latter having state-of-the-art performance. This work proposes an alternative algorithm (NestQuant) for quantizing weights, KV-cache, and activations. The

---

[*]Equal contribution  [1]Department of Electrical Engineering and Computer Science, Massachusetts Institute of Technology, Cambridge, MA [2]Independent [3]Rachel and Selim Benin School of Computer Science and Engineering, Hebrew University of Jerusalem, Israel. Correspondence to: Semyon Savkin <semyon@mit.edu>, Eitan Porat <ethan.porat@gmail.com>.

*Proceedings of the $42^{nd}$ International Conference on Machine Learning*, Vancouver, Canada. PMLR 267, 2025. Copyright 2025 by the author(s).

| Model | Bits ↓ | Bits (no zstd) ↓ | ARC-C ↑ | ARC-E ↑ | Hellaswag ↑ | PIQA ↑ | Winogrande ↑ | Zero-shot Avg ↑ | Wikitext2 ppl ↓ |
|---|---|---|---|---|---|---|---|---|---|
| Baseline (FP16) | 16 | 16 | 0.54 | 0.78 | 0.79 | 0.81 | 0.74 | 0.73 | 6.1 |
| **Weights only** | | | | | | | | | |
| LLM-QAT | 4.00 | - | 0.51 | 0.77 | 0.48 | 0.79 | 0.72 | 0.65 | 7.7 |
| GPTQ | 4.00 | - | 0.47 | 0.72 | 0.74 | 0.77 | 0.71 | 0.68 | 7.2 |
| SpinQuant | 4.00 | - | **0.54** | 0.77 | 0.78 | **0.80** | 0.72 | **0.72** | 6.5 |
| NestQuant $q = 14, k = 4$ (ours) | **3.99** | 4.06 | 0.53 | **0.78** | **0.79** | **0.80** | **0.73** | **0.72** | **6.3** |
| **Weights + KV cache** | | | | | | | | | |
| SpinQuant | 4.00 | - | 0.51 | 0.77 | 0.77 | 0.78 | 0.69 | 0.70 | 6.6 |
| NestQuant $q = 14, k = 4$ (ours) | **3.99** | 4.06 | **0.53** | **0.78** | **0.79** | **0.79** | **0.74** | **0.72** | **6.4** |
| **Weights, KV cache, activations** | | | | | | | | | |
| LLM-QAT | 4.00 | - | 0.27 | 0.41 | 0.38 | 0.60 | 0.53 | 0.44 | 52.5 |
| Quarot | 4.00 | - | 0.44 | 0.67 | 0.75 | 0.75 | 0.66 | 0.67 | 8.4 |
| SpinQuant | 4.00 | - | **0.51** | **0.75** | 0.75 | 0.77 | 0.66 | 0.68 | 7.3 |
| NestQuant $q = 14, k = 4$ (ours) | **3.99** | 4.06 | **0.51** | **0.75** | **0.78** | **0.79** | **0.72** | **0.71** | **6.6** |

*Table 1.* 4-bit quantization of Llama-3-8B. The bits column for NestQuant corresponds to actually measured average number of bits per entry (when a vector of auxiliary scaling coefficients $\beta$ is compressed via zstd) and the second column shows quantization rate when no compression step is used.

algorithm is motivated by recent theoretical work on approximate matrix multiplication and follows several classical ideas such as Conway and Sloane's Voronoi codes and their algorithms for finding the closest lattice vector in some canonical lattices in low/moderate dimensions.

Throughout this paper, NestQuant is described with the 8-dimensional Gosset lattice, but the framework is general and can be similarly implemented using any other base lattice, in any dimension, provided that a low-complexity closest lattice vector algorithm exists for this lattice.

### 1.1. Summary of results

The NestQuant algorithm is described in Section 4. NestQuant is a generic drop-in replacement for any matrix multiplication. Its performance for synthetic random Gaussian matrices comes pretty close to information-theoretic limits (see Fig. 3) and significantly outperforms uniform quantization employed by SpinQuant. Switching from a scalar (uniform) quantization to vector quantization requires some price to pay computationally (Section E.1), however, among vector quantizers NestQuant is rather economical as it builds upon the Voronoi Codes framework of (Conway & Sloane, 1983). We mention that in the presence of activation quantization, it is important to quantize weights properly – an innovation we call QA-LDLQ (Section 4.5).

Applying NestQuant to quantizing an actual LLMs (Llama 2 and 3 with parameter count ranging from 1B to 70B) shows massive end-to-end improvement: Fig. 1 shows a significant reduction of perplexity compared to the previous methods; Table 1 confirm enhanced performance on standard LLM benchmarks; and Table 2 show NestQuant stays consistently on top for different model sizes. In fact, NestQuant in full quantization (weights, activations, and KV cache) outperforms SOTA results with quantized weights and activations (but not KV cache).

The main source of improvement of NestQuant is demonstrated in Fig. 2 (although NestQuant uses an 8-dimensional Gosset lattice, not a 2D hexagonal one). More details on this as well as directions for improvement are discussed in Section 3.

Thus, we believe that NestQuant offers an excellent alternative to other algorithms. It quantizes weights, KV-cache and activations, and achieves significant improvement on both synthetic and real data.

### 1.2. Paper organization

We start with a detailed review of classical work on vector quantization and modern LLM quantization (Section 2). Then in Section 3 we explain the motivation for each step of the algorithm. Section 4 contains the pseudocode of the algorithm and diagram of quantized LLM. Finally, Section 5 concludes with details about empirical performance. Further details and evaluations are relegated to the Appendices.

## 2. Prior work

We briefly survey prior work, which we separate into work by information theorists and by the ML community.

### 2.1. Information-theoretic quantization

Rate $R$ quantization of an information source $X$ in $\mathbb{R}^n$ is the operation of encoding it to $nR$ bits, from which a decoder can produce a reconstruction $\hat{X} \in \mathbb{R}^n$ that has small *distortion* with respect to $X$. The most popular distortion criterion is the quadratic loss, where the expected distortion is defined as $D = \frac{1}{n}\mathbb{E}\|X - \hat{X}\|^2$, and here we restrict attention to this loss. Characterization of the optimal tradeoff between $R$ and $D$ is a classic topic in information theory, e.g. (Polyanskiy & Wu, 2024, Part V).

For a Gaussian source $X \sim \mathcal{N}(0, I_n)$ the rate-distortion

theorem states that any compressor at rate $R$ must satisfy $D \geq D(R) \triangleq 2^{-2R}$. Furthermore, as dimension $n$ increases there exist quantizers with distortion approaching $D(R)$. Notably, such quantizers can be made universal, in the sense that they attain distortion $D(R)$ not only for iid Gaussian $X$ but for any (even adversarial) input as long as its Euclidean norm is $(1 + o(1))\sqrt{n}$.

One way for constructing these universal quantizers is based on lattices (Zamir, 2014) that admit much more structure than more classical random codes (and $\epsilon$-nets).

Arguably, the most notable lattice-based quantization scheme is the family of Voronoi codes (Conway & Sloane, 1983), which we use in this work.

How does one convert a quantizer adapted to Gaussian inputs to work (with the same guaranteed loss) on non-Gaussian data? In a nutshell, the idea is simple: if $U$ is chosen to be a random $n \times n$ orthogonal matrix then the entries of $UX$ will be distributed as iid Gaussian (Stam, 1982). This idea of applying random rotations to smooth out the distribution of the quantizer's input may be viewed as a special case of high-dimensional companding (Gersho, 1979), and has been been used in various applications, such as image compression (Hung & Meng, 1998), and as a potential replacement for dithering (Hadad & Erez, 2016), to name a few.

In the context of LLMs, the goal in quantization is slightly different since quantization is used to facilitate approximate matrix multiplication with reduced burden on the memory bandwidth. For example, when quantizing two vectors $X, Y \in \mathbb{R}^n$ the goal *is not* to approximate them but to approximate their inner product. Recently, information-theoretic characterization of this task was completed in (Ordentlich & Polyanskiy, 2024). Specifically, the authors show that if $X, Y \sim \mathcal{N}(0, I_n)$ are independent then for any algorithm operating on the basis of rate-$R$ quantized representations of $X$ and $Y$ we must have

$$\mathbb{E}(X^\top Y - \widehat{X^\top Y})^2 \geq n\Gamma(R), \qquad (1)$$

where $\widehat{X^\top Y}$ is the reconstructed inner product, and

$$\Gamma(R) = \begin{cases} 1 - \left(1 - \left(2 \cdot 2^{-2R^*} - 2^{-4R^*}\right)\right)\frac{R}{R^*} & R < R^* \\ 2 \cdot 2^{-2R} - 2^{-4R} & R \geq R^* \end{cases}, \qquad (2)$$

where $R^* \approx 0.906$ is a solution to a certain transcendental fixed-point equation.

The same paper also constructs *universal quantizers* based on nested lattices that asymptotically (as $n \to \infty$) achieve this lower bound. Note that extension from vectors to matrices can be made trivially by observing that one can quantize each column separately and treat matrix product as a collection of inner products.

*Remark* 2.1. Using optimal quantizers, in terms of achieving $D(R)$, for quantizing $X$ and $Y$, and then computing the inner product of the quantized vectors (maybe with some MMSE scaling) will not necessarily lead to the optimal performance $\Gamma(R)$. The reason is that the covariance matrix of each vector's quantization error affects the inner product distortion through its trace, and *its Frobenius norm* (Ordentlich & Polyanskiy, 2024). For optimal lattice quantizers in any dimension, the Frobenius norm is guaranteed to be minimal (Zamir & Feder, 1996), and consequently, using lattice quantizers seems essential for achieving $\Gamma(R)$.

In this work we show that with appropriate tweaks Voronoi codes can indeed result in practical fast and efficient algorithms for LLM quantization. We emphasize that most of our work is on simply developing a drop-in replacement for quantized matrix product and as such is not specific to LLMs.

The idea of applying random rotations to "Gaussianize" inputs in the context of approximate inner product computation is even more natural than in the standard quantization. Indeed, since one is only interested in the inner product, not vectors themselves, one does not need to store (even a seed used to generate the) random orthogonal matrix. This has been long exploited in locality-sensitive hashing (LSH) algorithms,[1] which can be viewed as an (extremely low rate) quantizers for inner product approximation (Charikar, 2002; Datar et al., 2004; Andoni & Indyk, 2008). Unsurprisingly, as we will see next, random rotations have also been found quite useful for quantizing LLMs.

### 2.2. LLM quantization

One of the directions of prior research on LLM quantization is addressing the issue of activation outliers that hinder the quantization quality. These outliers are present in certain dimensions of activations, weights and KV cache. In LLM.int8() of (Dettmers et al., 2022), these outlier dimension are kept unquantized. In SmoothQuant (Xiao et al., 2023) authors balance the scale of outliers between weights and activations by modifying LayerNorm's diagonal matrices.

Going to random rotations, by rewriting matrix product $AB = (AU)(U^\top B)$ for an orthogonal matrix $U$, one gets matrices with much more Gaussian entries (few outliers) and can apply standard quantization algorithms. Some of the multiplications by $U$ can be merged with the weights (i.e. do not require additional runtime FLOPs), while the rest are applied at runtime. For the latter, matrices $U$ should have

---

[1]In LSH, one typically performs several random *projections* of the vector and quantizes them. This is equivalent to performing random rotation and quantizing only a small number of entries of the rotated vector.

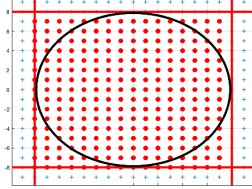 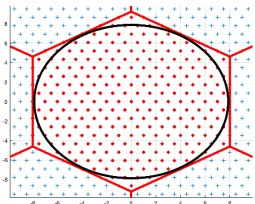

*Figure 2.* Demonstrating advantage of NestQuant in 2D. Typical weights and activations are vectors inside the black circle. Uniform quantization wastes about 32% of allocated bitstrings for vectors outside of the circle, while nested hexagonal lattices only wastes 15% (explicitly enumerating points inside the circle to avoid the waste is too slow to do at runtime). This allows NestQuant to use finer grid while quantizing to the same rate $R$. The gain becomes much more dramatic in higher dimensions.

structure to enable fast multiplication. For example, QuaRot (Ashkboos et al., 2024) uses randomized Hadamard matrices as coordinate transformations, which can be applied to a vector of size $n$ in $O(n \log n)$ additions. SpinQuant (Liu et al., 2024) uses a rotation parametrization with four orthogonal matrices $R_1$, $R_2$, $R_3$, $R_4$, where $R_1$ and $R_2$ can be arbitrary orthonormal matrices, and $R_3$ and $R_4$ should have a fast multiplication algorithm. The authors use Cayley SGD (Li et al., 2020) to optimize $R_1$ and $R_2$ for minimization of the quantization error, while the matrices $R_3$ and $R_4$ are chosen to be random Hadamard.

Starting from LLM.int8() most of the schemes used uniform quantization (i.e. where a floating point vector simply rounded to a nearest integer after an appropriate rescaling). To the best of our knowledge, so far non-scalar quantization has only been used for weight-only compression for LLMs in QuIP# (Tseng et al., 2024a), which uses E8P codebook for 2-bit quantization, and applies Resdidual Vector Quantization (Juang & Gray, 1982) to get a 4-bit quantization scheme; and QTIP (Tseng et al., 2024b) which uses trelis codebook. Unfortunately these methods appear to be too expensive to apply in runtime, perhaps explaining why non-uniform quantization for activations and KV-cache was not attempted before this work.

Finally, when quantizing weight matrices, one may notice that MSE distortion loss should be replaced by a weighted-MSE loss dependent on the statistics of the incoming activations. We refer to this type of algorithms as LDLQ, following authors of QuIP (Chee et al., 2023), QuIP# (Tseng et al., 2024a) and GPTQ (Frantar et al., 2022). In the presence of activation quantization, however, the LDLQ needs to be modified (see Section 4.5 introducing QA-LDLQ). Equipping NestQuant with QA-LDLQ incurs significant improvement; see results in Section 5.

## 3. Outline of NestQuant approach

In this section we outline the main components of our approach. A detailed description is brought in the following section.

When designing a quantizer, one needs to make some assumptions about the distribution of the source that will be fed to it. While weights (and sometimes activations) can be well-approximated by Gaussians, their magnitudes are wildly varied. Thus, one employs two ideas: normalization and random rotation.

**Normalization:** In most of the literature, the normalization is done by taking an input vector of large dimension $n$ (e.g. $n = 4096$ for Llama-3), dividing by the $L_\infty$ norm to get entries to be in $[-1, 1]$ and then applying uniform quantization. This is suboptimal for two reasons: one is that uniform quantization induces error that is distributed uniformly on the small cube, which is suboptimal from the MSE point of view. Second reason, much more serious, is known as the shaping gain and demonstrated on Fig. 2. When entries of the vector are Gaussian, it will typically lie inside the black circle. Thus those grid elements outside of it will almost never be used, wasting bitspace.

Instead, we use normalization by the $L_2$-norm (see Algorithm 3) and then use points inside the Voronoi region of a Gosset lattice, which as Fig. 2 (right) demonstrates wastes a lot fewer bitstrings for rare vectors, thus allowing us to use finer grids.

**Random rotation:** When input to the quantizer significantly differs from the presumed model (of iid Gaussians), performance can become quite poor. As discussed previously, multiplying by a random orthoghonal matrix $U$ provides an excellent fix. Specifically, $UX$ vector becomes uniform on the $n$-sphere of radius $\sqrt{n}$, and small chunks of this vector have distribution very close to Gaussian iid. In particular, the total variation between any subset of $d$ coordinates and $\mathcal{N}(0, I_d)$ is $O(d^2/n)$ (Stam, 1982), such that for $d = o(\sqrt{n})$ what we quantize is effectively iid Gaussian.

**Lattices:** A lattice $\Lambda \subset \mathbb{R}^d$ is a discrete subgroup of $\mathbb{R}^d$. Any lattice $\Lambda \subset \mathbb{R}^d$ has a (non-unique) generating matrix $G \in \mathbb{R}^{d \times d}$, such that $\Lambda = G\mathbb{Z}^d$ (that is, $\Lambda$ is the integer span of the columns of $G$). We define the nearest neighbor quantizer $Q_\Lambda : \mathbb{R}^d \to \Lambda$ as

$$Q_\Lambda(x) = \operatorname*{argmin}_{\lambda \in \Lambda} \|x - \lambda\|, \tag{3}$$

where ties are broken arbitrarily, but in a systematic manner. The Voronoi region $\mathcal{V}_\Lambda$ is defined as the set of all points in $\mathbb{R}^d$ that are closer to $0$ than to any other lattice point

$$\mathcal{V}_\Lambda = \left\{ x \in \mathbb{R}^d \; : \; Q_\Lambda(x) = 0 \right\}. \tag{4}$$

The covolume of a lattice is defined as $\operatorname{covol}(\Lambda) = |\det G| = \operatorname{vol}(\mathcal{V}_\Lambda)$. We say that a lattice $\Lambda_c \subset \mathbb{R}^d$ is *nested*

in the lattice $\Lambda_f \subset \mathbb{R}^d$, if $\Lambda_c \subset \Lambda_f$. Note that for any integer $q \geq 2$ we have that $q\Lambda \subset \Lambda$, and that $\Lambda/q\Lambda \cong (\mathbb{Z}/q\mathbb{Z})^d$. For an introduction to lattices and their role in quantization, see (Zamir, 2014).

**Complexity of lattice quantization:** In order to explain our choice of nested lattice quantizer, we need to carefully balance several requirements. One of the key ones is complexity. It is known that finding (even approximating) a nearest lattice point is a well-known cryptographic assumption (Dinur et al., 2003). Thus, we are not suggesting to directly operate on $n$-dimensional lattices. Instead, we partition the $n$-vector into sections, each of dimension $d$ and apply lattice quantization to $d$-subvectors. Equivalently, our vector quantizers for $\mathbb{R}^n$ are constructed as Cartesian products of vector quantizers of small dimension $d$ (we will take $d = 8$ for all experiments).

**Granular and overload quantization errors:** There are two different sources of errors for lattice quantizers. The first is called *granular* quantization error, and is related to the second moment of the lattice Voronoi region. A common way to measure the granular error corresponding to a lattice $\Lambda \subset \mathbb{R}^d$ is via the normalized second moment (NSM) defined as

$$G(\Lambda) = \frac{1}{\text{vol}(\mathcal{V}_\Lambda)^{1+\frac{2}{d}}} \frac{1}{d} \int_{x \in \mathcal{V}_\Lambda} \|x\|^2 dx. \qquad (5)$$

This quantity corresponds to the MSE when $\Lambda$ is normalized to have unit covolume and is then used as a quantizer. It is well known that the optimal (smallest) NSM among all lattices in $\mathbb{R}^d$ approaches $\frac{1}{2\pi e}$ from above as $d$ increases (Zamir, 2014). Furthermore, for $d = 1$ we get $G(\mathbb{Z}) = \frac{1}{12}$. Consequently, in terms of granular error, by using high-dimensional lattices instead of the simple scalar quantizer based on $\mathbb{Z}$ we can already gain a factor of $\frac{2\pi e}{12} \approx 1.42329$ in performance (the Gosset lattice achieves 1.16 gain).

Notice, however, that representing $x$ as $Q_\Lambda(x)$, the nearest lattice point in $\Lambda$, requires infinitely many bits, since the lattice is infinite. Since we only have $2^{dR}$ bitstrings to allocate, we need to select a subset of $\Lambda$ that will be actually used. Selection of a body $\mathcal{S} \subset \mathbb{R}^d$ so that $|\Lambda \cap \mathcal{S}| = 2^{dR}$ is called *shaping*. If $Q_\Lambda(x) \in \mathcal{S}$, then the quantization error $x - Q_\Lambda(x)$ will be in $\mathcal{V}_\Lambda$ and we will only suffer from a granular error. However, when $Q_\Lambda(x) \notin \mathcal{S}$ the quantization error is no longer in $\mathcal{V}_\Lambda$ and may be have much greater magnitude than a typical granular error. We refer to those type of errors, where $Q_\Lambda(x) \notin \mathcal{S}$, as *overload* errors.

Generally speaking, in order to achieve a small quantization error, one must keep the probability of overload very small. This can be attained by scaling up the codebook to $\beta\mathcal{C} = \beta\Lambda \cap \beta\mathcal{S}$ with a large enough $\beta > 0$ such that overload becomes very rare. However, increasing $\beta$ also increases the squared granular error by a factor of $\beta^2$. Thus, one

would like to use the smallest possible $\beta$ for which overload is rare. In order to allow for smaller $\beta$, we would like to choose $\mathcal{S} \subset \mathbb{R}^n$ such that $\beta\mathcal{S}$ captures as much Gaussian mass as possible.

Denote by $\mu = \mathcal{N}(0, I_d)$ the standard Gaussian measure. Since we need $2^{dR} = |\Lambda \cap \mathcal{S}| \approx \frac{\text{vol}(\mathcal{S})}{\text{covol}(\Lambda)}$, a good shaping region $\mathcal{S}$ maximizes $\mu(\mathcal{S})$, which in turn minimizes the overload probability that is approximated by $1 - \mu(\mathcal{S})$, under a volume constraint. Clearly, the optimal $\mathcal{S}$ under this criterion is $r\mathcal{B}$ where $\mathcal{B} = \{x \in \mathbb{R}^d : \|x\| \leq r_{\text{eff}}(1)\}$ is a Euclidean ball with radius $r_{\text{eff}}(1)$ chosen such that $\text{vol}(\mathcal{B}) = 1$, and $r$ is chosen such that $\text{vol}(r\mathcal{B}) = r^d$ satisfies the required volume constraint. Unfortunately, for $d > 1$ the codebook $\mathcal{C} = \Lambda \cap r\mathcal{B}$ loses much of the lattice structure, and does not admit an efficient enumeration, and consequently encoding and decoding require using a lookup table (LUT). QuIP# used this approach with $\Lambda = E_8$ (same as we do) and $\mathcal{S} = r\mathcal{B}$. However, this seems to only be possible for quantizing weights and not activations as complexity makes runtime implementation too slow.[2]

**Using int8-multipliers:** One often mentioned advantage of uniform quantization compared to other approaches is the fact that it approximates any matrix as a product of diagonal matrix (of norms) and an integer matrix. Thus, during multiplication one can leverage faster int-cores rather than floating-point multiplication. Note that if there exists a scaling coefficient $\alpha > 0$ such that $\alpha\Lambda \subset \mathbb{Z}^d$, then one can still use int-multipliers even for lattice-quantized vectors.

**Voronoi codes/nested lattice codes:** In Voronoi codes (Conway & Sloane, 1983) the same lattice $\Lambda$ is used for both quantization and shaping. In particular, the shaping region is taken as $\mathcal{S} = q\mathcal{V}_\Lambda$, where $q = 2^R$ is an integer. As elaborated below, if $Q_\Lambda(x)$ admits an efficient implementation, one can efficiently perform encoding and decoding to the codebook $\mathcal{C} = \Lambda \cap (2^R\mathcal{V}_\Lambda) \cong \Lambda/2^R\Lambda$. Moreover, in stark contrast to ball-based shaping, the encoding and decoding complexity does not depend on $R$.

*In summary,* a good choice of lattice $\Lambda$ should therefore have: 1) efficient lattice decoding algorithm; 2) small NSM; 3) large $\mu(\mathcal{V}_\Lambda)$; 4) be a subset of standard integer lattice $\mathbb{Z}^d$.

In this work, we use the Gosset lattice ($E_8$) that satisfies all these properties. It has a fast decoding algorithm (Algorithm 5), its NSM is $\approx 0.0716821 \approx 1.2243\frac{1}{2\pi e}$ (Agrell & Allen, 2023), and its Gaussian mass $\mu(r\mathcal{V}_{E_8})$ is very close to $\mu(r\mathcal{B})$ (the Gosset lattice has unit covolume, so $\text{vol}(r\mathcal{V}_{E_8}) = \text{vol}(r\mathcal{B})$). The last point is illustrated in

---

[2]We note that QuIP# cleverly exploits symmetries of $E_8$ to show that an $R = 2$ bit quantizer can be implemented using an LUT of size $2^{d\frac{R}{2}} = 2^8$, but we believe this is still too slow, and furthermore does not naturally extend to different quantization rates.

Figure 5, where the large loss for cubic shaping with respect to lattice shaping is also evident. The gap between Voronoi/Ball shaping and cubic shaping becomes much more significant as the dimension increases. This follows since for large $d$ we have that for $X \sim \mu = \mu_d$ the $\ell_\infty$ norm $\|X\|_\infty$ concentrates around $\sqrt{2 \ln d}$. Thus, for $r < 2\sqrt{2 \ln d}$ we have that $\mu(r\text{CUBE}) \to 0$, whereas for any $r > \frac{\sqrt{d}}{r_{\text{eff}}(1)} = \sqrt{2\pi e}(1+o(1))$ we have that $\mu(r\mathcal{B}) \to 1$. Note that SpinQuant uses high-dimensional cubic shaping, and therefore its MSE distortion suffers a $O(\ln d)$ multiplicative gap with respect to the optimal distortion.

**Overload avoidance via union of Voronoi codes:** Because we rely on lattice quantizers of relatively small dimension ($d = 8$), even if $\mu(r\mathcal{V}_\Lambda)$ is very close to $\mu(r\mathcal{B})$, overload events are unavoidable. This follows because in small dimension the norm of a iid Gaussian vector is not sufficiently concentrated. Thus, if one is restricted to $\mathcal{C} = \beta(\Lambda \cap (2^R \mathcal{V}_\Lambda))$ the parameter $\beta$ must be taken quite large in order to keep the overload probability small. This in turn, incurs a significant penalty in the obtained distortion. As a remedy, rather than using a Voronoi code, we take $\mathcal{C}$ as a union of (a small number) of Voronoi codes in different scales. Namely, we take $\mathcal{C} = \cup_{t=1}^{k} \beta_t(\Lambda \cap (2^R \mathcal{V}_\Lambda))$, where $\beta_1 < \cdots < \beta_k$. The smallest values of $\beta_t$ are set such that overload is not too common but not extremely rare, such that for most realizations of a Gaussian vector $X \in \mathbb{R}^d$ the distortion is close to the fundamental limit $D(R)$. Whenever $X$ is atypically large, there will be overload in $\beta_t(\Lambda \cap (2^R \mathcal{V}_\Lambda))$ for small $t$, but not for large $t$, such that the quantization error will be in $\beta_t \mathcal{V}_\Lambda$ for one of the larger values of $\{\beta_t\}$.

The details of choosing $k$ and values of $\beta_t$ are described in Section F. Here we only note that the overall compression rate becomes $R + \frac{1}{d}\log_2 k$. In some cases, we are using nvcomp (NVIDIA Corporation, 2025) for compressing a vector of $n/8$ chosen betas, in which case rate penalty is reduced below $\frac{1}{d}\log_2 k$. We note that in our comparisons, including Table 1, we always use this effective rate for a fair comparison with other algorithms.

**NestQuant, SpinQuant and theory:** As mentioned above, the use of nested lattice codes is rooted in theory. In (Ordentlich & Polyanskiy, 2024) it was shown that nested lattice quantizers of high-dimensions attain the optimal rate-distortion tradeoff for matrix multiplication. Since the lattices used for proving that result do not admit efficient lattice decoding, here we resort to $n$-dimensional lattices constructed as the Cartesian product of $n/d$ copies of the Gosset lattice, whose dimension is $d = 8$. To understand how much loss in efficiency this leads to, Fig. 3 compares NestQuant, SpinQuant (uniform quantization with cubic shaping) and information-theoretic lower bound (1), for synthetic data. Details of this experiment can be found in Section 5.1. We can see that NestQuant is reasonably close to the fundamen-

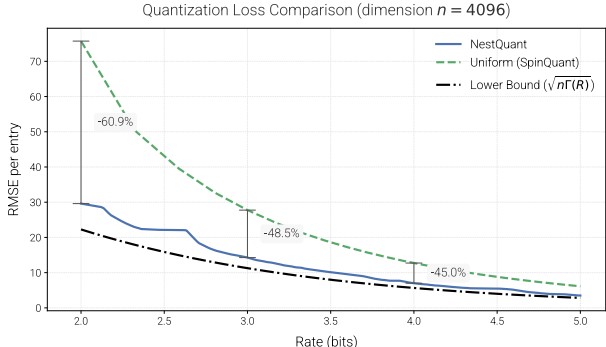

*Figure 3.* RMSE for quantized matrix multiplication for iid $\mathcal{N}(0,1)$ matrices. NestQuant algo is optimized over $q$ and multiple $\beta$'s. Also shown is information-theoretic lower bound from (1).

tal limit and significantly outperforms SpinQuant.

## 4. Detailed Method

### 4.1. Nested lattice codebook

In this section, we describe the construction for a Vector Quantization (VQ) codebook of size $q^d$ for quantizing a $d$-dimensional vector, where $q$ is an integer parameter. This construction is based on Voronoi codes (Conway & Sloane, 1983) and admits efficient encoding and decoding algorithms, whenever the base lattice has an efficient closest lattice vector algorithm. Another appealing feature of Voronoi codes is that the encoding/decoding complexity is independent of the quantization rate $R = \log_2(q)$.

**Definition 4.1** (Voronoi code (Conway & Sloane, 1983)). The Voronoi codebook with base lattice $\Lambda \subset \mathbb{R}^d$ and nesting ratio $q \in \mathbb{N}$, corresponding to rate $R = \log_2(q)$ bits/entry, is defined as $C = \Lambda \cap \mathcal{V}_{q\Lambda} = \Lambda \cap (q\mathcal{V}_\Lambda) \subset \mathbb{R}^d$. In particular $\lambda \in \Lambda$ belongs to codebook $C$ iff $\lambda \in \mathcal{V}_{q\Lambda}$, and $|C| = q^d$.

The Voronoi code consists of the minimum energy representative of each coset in $\Lambda/q\Lambda \cong (\mathbb{Z}/q\mathbb{Z})^d$. Consequently, we can represent each coset, and hence, each codeword in $C$, as an element of $\mathbb{Z}_q^d$ (Conway & Sloane, 1983).

Assuming we have access to an oracle $Q_\Lambda(x)$, which maps $x \in \mathbb{R}^d$ to its closest point in $\Lambda$, the quantization (encoding) and dequantization (decoding) of a Voronoi code are described in Algorithm 1 and Algorithm 2, respectively. Here, $G \in \mathbb{R}^{d \times d}$ is a generator matrix of $\Lambda$. The encoder first maps $x \in \mathbb{R}^d$ to its nearest lattice point $p = Q_\Lambda(x)$. Since it only has a budget of $dR$ bits for describing $p$, it only describes the coset of $\Lambda/q\Lambda$ it belongs to. This is done by sending $v \mod q$, where $v \in \mathbb{Z}^d$ is the integer vector for which $p = Gv$, referred to as *the coordinates of* $p$. Upon receiving $v \mod q$, the decoder knows that $Q_\Lambda(x) \in p + q\Lambda$, and has to choose one point in this coset as the reconstruc-

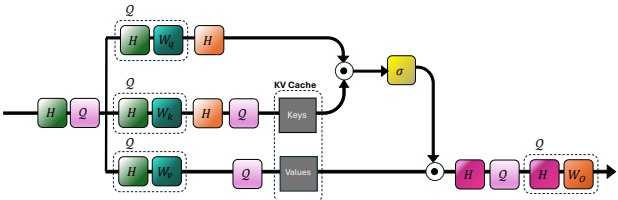

*Figure 4.* The quantization scheme of multi-head attention. $H$ is Hadamard rotation described in 4.3. $\mathcal{Q}$ is the quantization function described in 4.2

tion $\hat{x} \in \mathbb{R}^d$. It chooses $\hat{x}$ as the minimum energy vector in $p + q\Lambda$, corresponding to the unique point in $p + q\Lambda \cap \mathcal{V}_{q\Lambda}$. We have that $\hat{x} = Q_\Lambda(x)$ iff $Q_\Lambda(x) \in \mathcal{V}_{q\Lambda} = q\mathcal{V}_\Lambda$. When $Q_\Lambda(x) \notin q\mathcal{V}_\Lambda$ the quantizer is in *overload*.

---

**Algorithm 1** Encode

    **Input:** $x \in \mathbb{R}^d$, $Q_\Lambda$
    $p \leftarrow Q_\Lambda(x)$
    $v \leftarrow G^{-1}p$                 ▷ coordinates of $p$
    **return** $v \bmod q$     ▷ quantized representation of $p$

---

**Algorithm 2** Decode

    **Input:** $c \in \mathbb{Z}_q^d$, $Q_\Lambda$
    $p \leftarrow Gc$           ▷ equivalent to answer modulo $q\Lambda$
    **return** $p - q\,Q_\Lambda\!\left(\frac{p}{q}\right)$

---

In our experiments for this paper, we used the Gosset ($E_8$) lattice as $\Lambda$ with $d = 8$. This lattice is a union of $D_8$ and $D_8 + \frac{1}{2}$, where $D_8$ contains elements of $\mathbb{Z}^8$ with even sum of coordinates. There is a simple algorithm for finding the closest point in the Gosset lattice, first described in (Conway & Sloane, 1982). We provide the pseudocode for this algorithm together with the estimation of its runtime in Appendix C.

### 4.2. Matrix quantization

When quantizing a matrix, we normalize its rows, and quantize each block of $d$ entries using the codebook. The algorithm 3 describes the quantization procedure for each row of the matrix.

We can take dot products of quantized vectors without complete dequantization using algorithm 4. We use it in the generation stage on linear layers and for querying the KV cache.

### 4.3. LLM quantization

Recall that we apply a rotation matrix $H$ to every weight-activation pair of a linear layer without changing the output

---

**Algorithm 3** NestQuant

    **Input:** $A$ — a vector of size $n = db$, $q$, array of $k$ scaling coefficients $\beta_1, \ldots, \beta_k$
    $QA$ — $n$ integers in $\{0, 1 \ldots, q - 1\}$     ▷ quantized representation
    $B$ — $b$ integers in $\{1, \ldots, k\}$     ▷ scaling coefficient indices
    $s \leftarrow \|A_i\|_2$             ▷ normalization coefficient
    $A \leftarrow \frac{A\sqrt{n}}{s}$
    **for** $j = 0$ **to** $b - 1$ **do**
        $err = \infty$
        **for** $p = 1$ **to** $k$ **do**
            $v \leftarrow A[dj + 1..dj + d]$
            $enc \leftarrow \text{Encode}\left(\frac{v}{\beta_p}\right)$
            $recon \leftarrow \text{Decode}(enc) \cdot \beta_p$
            **if** $err > |recon - v|_2^2$ **then**
                $err \leftarrow |recon - v|_2^2$
                $QA[dj + 1..dj + d] \leftarrow enc$
                $B_j \leftarrow p$
            **end if**
        **end for**
    **end for**
    **Output:** $QA$, $B$, $s$

---

**Algorithm 4** Dot product

    **Input:** $QA_1$, $B_1$, $s_1$ and $QA_2$, $B_2$, $s_2$ — representations of two vectors of size $n = db$ from Algorithm 3, array $\beta$
    $ans \leftarrow 0$
    **for** $j = 0$ **to** $b - 1$ **do**
        $p_1 \leftarrow \text{Decode}(QA_1[dj + 1..dj + d])$
        $p_2 \leftarrow \text{Decode}(QA_2[dj + 1..dj + d])$
        $ans \leftarrow ans + (p_1 \cdot p_2)\beta_{B_1[j]}\beta_{B_2[j]}$
    **end for**
    **return** $ans$

---

of the network. Let $n$ be the number of input features to the layer. Following (Tseng et al., 2024a; Liu et al., 2024; Lin et al., 2024):

- If $n = 2^k$, we set $H$ to be Hadamard matrix obtained by Sylvester's construction

- Otherwise, we decompose $n = 2^k m$, such that $m$ is small and there exists a Hadamard matrix $H_1$ of size $m$. We construct Hadamard matrix $H_2$ of size $2^k$ using Sylvester's construction, and set $U = H_1 \otimes H_2$.

Note that it's possible to multiply an $r \times n$ matrix by $H$ in $O(rn \log n)$ in the first case and $O(rn(\log n + m))$ in the second case, which is negligible to other computational costs and can be done online.

In NestQuant, we quantize all weights, activations, keys, and values using Algorithm 3. We merge the Hadamard rotation with the weights and quantize them. We also apply the Hadamard rotation and quantization to the activations before linear layers. We also apply rotation to keys and queries, because it will not change the attention scores, and we quantize keys and values before putting them in the KV cache. Figure 4 illustrates the procedure for multi-head attention layers.

When quantizing a weight, we modify the NestQuant algorithm by introducing corrections to unquantized weights when a certain vector piece is quantized via the QA-LDLQ mechanism, described in section 4.5. It is based on LDLQ algorithm, described in section 4.1 of (Tseng et al., 2024a).

### 4.4. Optimal scaling coefficients

One of the important parts of the algorithm is finding the optimal set of $\beta_i$. Given the distribution of $d$-dimensional vectors that are quantized via a Voronoi codebook, it is possible to find an optimal set of given size exactly using a dynamic programming approach, which is described in Appendix F.

### 4.5. LDLQ with quantized activations (QA-LDLQ)

We note that in the presence of activation quantization, the LDLQ and GPTQ algorithms for weight quantization become significantly suboptimal [3], thus necessitating a correction (QA-LDLQ) that we describe here.

Let $W$ be the weight (shape $a \times n$), $X$ be a random original (unquantized) activation vector (shape $n \times 1$), and $Z$ be $X$'s quantization error, which we model as random zero-mean noise (shape $n \times 1$) independent of $X$. Then, if $U$ is the quantized weight, the output quantization error becomes $\delta(U) := WX - U(X+Z)$, so that the loss to be minimized is $\mathbb{E}[\|\delta(U)\|^2]$ instead of $\mathbb{E}[\|(W-U)X\|^2]$ that LDLQ and GPTQ minimize.

**Lemma 4.2.** *Suppose $Z$ is independent from $X$, $\mathbb{E}[Z] = 0$, and let $H = \mathbb{E}[XX^\top]$ and $J = \mathbb{E}[ZZ^\top]$. Then for any set $\mathcal{C}_Q \subset \mathbb{R}^{a \times n}$*

$$U^* = \operatorname*{argmin}_{U \in \mathcal{C}_Q} \mathbb{E}[\|\delta(U)\|^2]$$
$$= \operatorname*{argmin}_{U \in \mathcal{C}_Q} (\tilde{W} - U)(H + J)(\tilde{W} - U)^\top, \quad (6)$$

*where $\tilde{W} = WH(H + J)^{-1}$.*

Thus, QA-LDLQ consists of a) computing $\tilde{W}$; b) running the standard LDLQ but with matrix $\tilde{W}$ as input and Hessian set to $H + J$. More details, practical aspects of QA-LDLQ, as well as proof of the lemma are found in Appendix B.

---

[3] In fact, using original LDLQ on Llama-3-70B produces $\infty$ perplexity at W4A4 setting due to significant outliers in layer 0.

### 4.6. Algorithm summary

Here we describe the main steps of NestQuant.

1. Collect the statistics for LDLQ via calibration data. For each linear layer with in-dimension $d$, we compute a $d \times d$ "Hessian" matrix $H$.

2. We choose an initial set of scaling coefficients $\hat{\beta}$, and for each weight we simulate LDLQ quantization with these coefficients, getting a set of 8-dimensional vectors to quantize.

3. We run a dynamic programming algorithm described in Appendix F on the 8-vectors to find the optimal $\beta$-values for each weight matrix.

4. We also run the dynamic programming algorithm for activations, keys, and values for each layer. To get the distribution of 8-vectors, we run the model on a small set of examples.

5. We quantize the weights using QA-LDLQ and precomputed $\beta$.

6. During inference, we quantize any activation before it is passed to the linear layer, and any KV cache entry before it is saved.

Note that we do not undertake any expensive (but surely useful) fine-tuning, such as optimizing rotation matrices or post-quantization training, as in (Liu et al., 2024) and (Tseng et al., 2024a), since our goal is demonstrating the basic primitive, not obtaining the absolute SOTA numbers.

## 5. Experiments

### 5.1. Simulated Data

We compared the mean $L_2$ loss per entry of SpinQuant to the uniform $L_\infty$-scaling quantizer (used in SpinQuant and other methods). The mean $L_2$ loss per entry for the product of two matrices $A \in \mathbb{R}^{n \times k}, B \in \mathbb{R}^{m \times k}$ is computed as $\frac{\|AB^T - \hat{A}\hat{B}^T\|_2}{nm}$. We set $n = k = m = 4096$ and sampled two matrices $A, B$ with unit normal distribution $A_{ij}, B_{ij} \sim \mathcal{N}(0,1)$. We compare to the lower bound from (1).

For NestQuant, we do a grid search over $(q, k)$. For given $q$ and $k$, we find the best subset in $\frac{1}{2} \cdot \{1, 2, \ldots, 50\}$ of scaling coefficients $\beta$ of size $k$ using the algorithm from Appendix F. Then we calculate the expected bits per entry computed as $\log_2 q + \frac{1}{8} \sum_{i=1}^{k} p(\beta_i) \log_2 p(\beta_i)$ where $p(\beta_i)$ is the probability that the $i$'th beta is used in quantization. In Figure 3, we plot the efficient frontier of bits per entry vs root mean $L_2$ loss.

| Bits (W-A-KV) | Method | Llama-2-7B | Llama-2-13B | Llama-2-70B | Llama-3-8B | Llama-3-70B |
|---|---|---|---|---|---|---|
| 16-16-16 | Floating point | 5.47 | 4.88 | 3.32 | 6.14 | 2.86 |
| 4-16-16 | QuaRot | 5.60 | 5.00 | 3.41 | - | - |
| | QuIP# | 5.56 | 4.95 | **3.38** | - | - |
| | OstQuant | 5.64 | 4.94 | 3.41 | 6.53 | 3.19 |
| | NestQuant | **5.53** | **4.93** | **3.38** | **6.31** | **3.14** |
| | NestQuantM | 5.55 | 4.95 | - | 6.35 | - |
| 4-16-4 | NestQuant | 5.57 | 4.96 | 3.39 | 6.37 | 3.19 |
| | NestQuantM | 5.59 | 4.99 | - | 6.49 | - |
| 4-4-16 | SpinQuant | 5.9 | 5.2 | 3.8 | 7.1 | - |
| | OstQuant | 5.60 | 5.14 | 3.57 | 7.24 | 3.97 |
| | DuQuant | 6.08 | 5.33 | 3.76 | 8.06 | - |
| 4-4-4 | QuaRot | 6.10 | 5.40 | 3.79 | 8.16 | 6.66 |
| | SpinQuant | 5.9 | 5.3 | 3.8 | 7.3 | - |
| | OstQuant | 5.91 | 5.25 | 3.59 | 7.29 | 4.01 |
| | NestQuant | **5.67** | **5.03** | **3.49** | **6.63** | **3.61** |
| | NestQuantM | 5.73 | 5.07 | - | 6.82 | - |

*Table 2.* The wikitext2 perplexity at context window of 2048 for various quantization methods of Llama models.

## 5.2. Llama results

We quantize the Llama-3-8B model (Llama Team, AI @ Meta, 2024) using different values of $q$. We choose the number of scaling coefficients ($k$) to be equal to 4, the Section H.1 explains the rationale behind this choice. More details on the hyperparameter choice of the experiments are in Appendix G. For each experiment, we compute the number of bits per entry similar to Section 5.1, but for the setup of compressed $\beta$ indices, we run the zstd compression algorithm instead of using the entropy of the distribution. As our evaluation metric, we use the perplexity on the validation split of wikitext2 with context length 2048.

We also perform the evaluation of NestQuant on various zero-shot benchmarks: ARC-Easy and ARC-Challenge (Clark et al., 2018), Hellaswag (Zellers et al., 2019), (Bisk et al., 2020), and Winogrande (Sakaguchi et al., 2020). The results on 4-bit models with comparisons to other models are summarized in Table 1.

## 5.3. Comparison to other methods

In comparisons to other methods, we focus on 4-bit setup, choosing $q = 14$ and $k = 4$. We show the WikiText2 perplexity comparisons for multiple Llama models in table 2, and other benchmark comparisons (for Llama-3-8B) in table 1. The methods that we include in the table are SpinQuant (Liu et al., 2024), QuIP# (Tseng et al., 2024a), QuaRot (Ashkboos et al., 2024), DuQuant (Lin et al., 2024), and OstQuant (Hu et al., 2025). On the WikiText2 dataset, we computed the perplexity scores of the quantized models on context size 2048.

| q | Bits | Bits (no zstd) | W | W + KV | W + KV + A |
|---|---|---|---|---|---|
| 14 | 3.99 | 4.06 | 6.308 | 6.379 | 6.633 |
| 12 | 3.76 | 3.83 | 6.376 | 6.475 | 6.841 |
| 10 | 3.50 | 3.57 | 6.486 | 6.640 | 7.251 |
| 8 | 3.18 | 3.25 | 6.700 | 6.968 | 7.989 |

*Table 3.* Wikitext2 perplexity of NestQuant quantization of Llama-3-8B at different rates. The "bits" column is the bit rate per entry with zstd compression of scaling coefficients, and "bits (no zstd)" is the bit rate without compression. The "W", "W+KV", and "W+KV+A" describe the quantization regime (whether weights, KV cache, or activations are quantized). The perplexity of non-quantized model is 6.139

NestQuant consistently achieves better perplexity metrics across different models for both weight-only regime and full quantization. In fact, for all models we have tested, except Llama-2-7B, NestQuant with W4A4KV4 (4-bit weights, activations, and KV-cache) quantization even outperforms previous works with W4A4KV16. On W4KV4A4 (4-bit weights, KV-cache, and activations) quantization of Llama 3-8B we achieve a perplexity score of 6.6 compared to 7.3 in SpinQuant and OstQuant. Even without LDLQ, we achieve a perplexity score of 6.8, which is still better. Finally, NestQuant outperforms QuIP# for weight-only Llama-2 quantization.

In addition, we evaluate a simpler version of the algorithm that is easier to run in hardware that is called NestQuantM. More detailes on it can be found in section D.

We present ablation studies for our design choices in in Appendix H.

## Impact statement

This paper presents work whose goal is to advance the field of Machine Learning. There are many potential societal consequences of our work, none which we feel must be specifically highlighted here.

## Acknowledgments

We want to thank Nikita Lazarev and Shang Yang for important discussions that helped us to implement an efficient CUDA kernel, as well as Guangxuan Xiao for discussions about the algorithm.

This work was supported in part by the National Science Foundation under Grant No CCF-2131115 and the ISF under Grant 1641/21. We thank Foundry.ai and Lambda Labs for compute resources.

Research was supported, in part, by the United States Air Force Research Laboratory and the United States Air Force Artificial Intelligence Accelerator under Cooperative Agreement Number FA8750-19-2-1000. The views and conclusions contained in this document are those of the authors and should not be interpreted as representing the official policies, either expressed or implied, of the United States Air Force or the U.S. Government. The U.S. Government is authorized to reproduce and distribute reprints for Government purposes notwithstanding any copyright notation herein.

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

# A. Figures

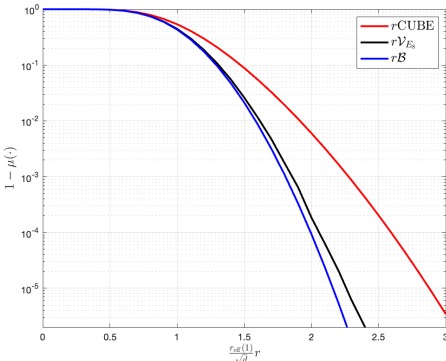

*Figure 5.* Complement Gaussian measure of a 8-dimensional cube (corresponding to shaping using an $\ell_\infty$ ball), a Voronoi region of the Gosset lattice $E_8$ (corresponding to shaping using Voronoi codes with base lattice $E_8$), and a Euclidean ball (corresponding to shaping with a ball, which does not admit efficient implementation)

# B. Details of QA-LDLQ

**LDLQ details.** We briefly mention the details of LDLQ (for more details we refer the reader to (Frantar et al., 2023) or (Chee et al., 2023)). The main observation is that we want to minimize $\|(W - U)X\|^2$ where $U$ is the quantized weight and $X$ is the matrix of activations. We observe that $\|(W - U)X\|^2 = \|(W - U)L_X \sqrt{D_X}\|^2$ where $L_X D_X L_X^\top = H = \mathbb{E}[XX^\top]$ and $L_X$ is a uni-lower-triangular matrix and $D_X$ is diagonal. LDLQ proceeds by optimally quantizing the last column of $U$ (for example using the quantizer for NestQuant). It then recursively removes the feedback $i$, $(W - U)(L_X - I)E_i$ from the rest of the columns (here $E_i$ is the zero matrix with the $ii$ index set to 1). This may be written compactly as $U = \mathcal{Q}(W - (W - U)(L_X - I))$ where $\mathcal{Q}$ quantizes each column separately.

**Motivation.** When quantizing Llama-3-70B with original LDLQ, we have noticed very poor (ppl $\sim 10^4$) performance. We have observed that the reason for this is quantization of activations for a small subset of linear layers (4 out of 560), and if they are left in full precision, the perplexity becomes more reasonable (less than 4). We have found them by choosing the layers with the largest *amplification ratio* — a concept we define next.

Consider a weight $W$ (shape $a \times n$), where $n$ is the number of in-features, and $a$ is the number of out-features. We define *amplification* of a random vector $X$ ($n \times 1$) by $W$ as $\alpha(W, X) := \frac{\mathbb{E}[\|WX\|_2]}{\mathbb{E}[\|X\|_2]}$. Then, if $X$ is the distribution of input activations, and $Z$ is a random Gaussian vector, we define the amplification ratio for $W$ as $\frac{\alpha(W, Z)}{\alpha(W, X)}$. We note that if a layer has large amplification ratio, its activations are harder to quantize, because the magnitude of quantization noise will be increased much more significantly than the magnitude of the activations themselves. One extreme example of this issue is the value projection of the attention of the first transformer block in Llama-3-70B. This layer has an amplification ratio of $\sim 157$, as computed from 10 wikitext2 sequences of length 2048. This makes naive 4-bit quantization of activations nearly impossible, as even small perturbations of the input of the layer are greatly amplified in the output.

We have developed QA-LDLQ to mitigate the issue of large amplification ratio. We model quantization noise as a random Gaussian vector with mean 0 and covariance matrix $J = \varepsilon^2 I$, where $\varepsilon^2$ depends on the quantization rate and the statistics of $X$. For large values of $\varepsilon^2$ the modified weight matrix $\tilde{W} = WH(H + \varepsilon^2 I)^{-1}$ is more robust to random perturbations of the input (thus, decreasing the amplification ratio), but on the other hand, large $\varepsilon^2$ results in a larger bias, as expressed in the term $C(W, H, J)$ below. Figure 6 demonstrates this tradeoff for the value projection layer mentioned earlier.

*Proof of Lemma 4.2.* Recalling the definition of $\tilde{W} = WH(H + J)^{-1}$, that $X$ and $Z$ are statistically independent, and $Z$ has zero mean, and that $H, J$ are positive semi-definite symmetric matrices, we have

$$\mathbb{E}\|\delta(U)\|^2 = \mathbb{E}\|(W - U)X - UZ\|^2 = (W - U)H(W - U)^\top + UJU^\top \tag{7}$$

$$= (\tilde{W} - U)(H + J)(\tilde{W} - U) + C(W, H, J), \tag{8}$$

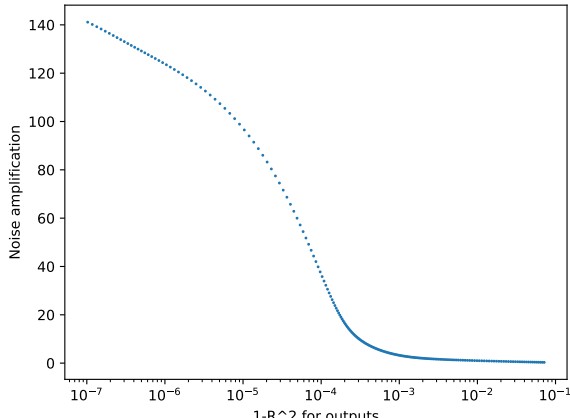 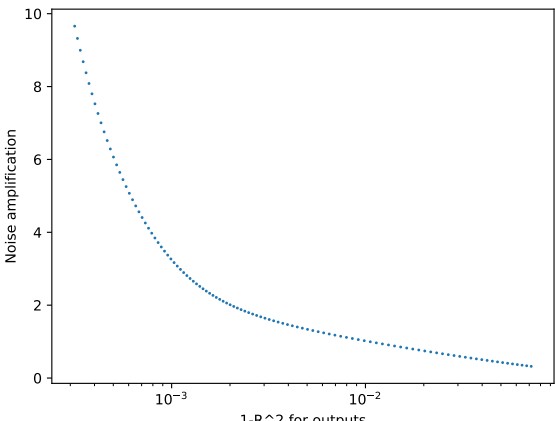

*Figure 6.* We run QA-LDLQ for value projection layer of the first transformer block of Llama-3-70B. We try different values of $\varepsilon$ on logarithmic scale from $10^{-5}$ to 1. For each $\varepsilon$, we find modified weight $\tilde{W}$, and plot the amplification ratio for $\tilde{W}$ in $y$-axis, as well as how close the outputs of the weight $\tilde{W}$ to the outputs of weight $W$. The value on $x$ axis is defined as $1 - R^2 := \frac{\mathbb{E}\|WX - \tilde{W}X\|^2}{\text{Var}(WX)}$, where $X$ contains activation inputs from 10 sequences of length 2048 from wikitext2. The right plot is bottom right corner of the left plot, zoomed in. We note that by paying a small price in the accuracy of the weight, we can reduce the amplification ratio dramatically.

where

$$C(W, H, J) = W(H - H(H + J)^{-1}H)W^\top, \tag{9}$$

is independent of $U$, and therefore does not affect the minimization. $\qquad\square$

## C. Gosset oracle

In this section, we discuss the algorithm for finding the nearest neighbour in $E_8$ lattice and estimate its performance in FLOPs (Floating Point Operations). We note that $E_8 = D_8 \cup D_8 + \frac{1}{2}$, where $D_8$ contains vectors in $\mathbb{Z}^8$ with even sum of coordinates. To compute $V_{E_8}(x)$, we compute two candidate points: $x_1 = V_{D_8}(x)$ and $x_2 = V_{D_8 + \frac{1}{2}}(x)$, and choose the one that has smaller $L^2$ distance to $x$.

To get $V_{D_8}(x)$, we can round each coordinate to the nearest integer. If the sum of rounded coordinates is odd, we need to "flip" the rounding direction of the coordinate for which the flip would cost the least. Note that finding the closest point in $V_{D_8 + \frac{1}{2}}$ works the same, but the rounding grid now contains half-integers, not integers.

In algorithm 5, we first round our vector down (getting $d$) and compute the mask ($g$) of whether it's optimal to round up for $D_8$. We note that the optimal rounding for $D_8 + \frac{1}{2}$ is $d + 0.5$, while the optimal rounding for $D_8$ is $d + g$.

We want to understand whether rounding to $D_8$ or $D_8 + \frac{1}{2}$ is better. Let $dist_i$ be the absolute distance from the $i$-th entry $x_i \in [d_i, d_i + 1]$ to the middle of this integer segment $d_i + 0.5 = x_{2,i}$. We note that the contribution of this point to the MSE for $D_8$ is $(0.5 - dist_i)^2$, while for $D_8 + \frac{1}{2}$ is $dist_i^2$. The difference is: $0.25 - dist_i + dist_i^2 - dist_i^2 = 0.25 - dist_i$. If the sum of this value over $i$ is negative (i.e. $\sum dist_i > 2$), it's optimal to quantize to $D_8$, otherwise to $D_8 + \frac{1}{2}$. In pseudocode, we store $\sum dist_i$ as $\Delta$

We note that we should check the constraint that the sum of coordinates in $D_8$ is even, and if it is not, "flip" one of the rounding directions. The optimal coordinate to flip can be determined through $dist$, and the new value of flipped coordinate — through $g$. We also need to update $\Delta$ given that the MSE difference changes.

The full pseudocode of the algorithm is in Algorithm 5.

---

**Algorithm 5** Oracle for the Gosset lattice

---

1: **Input:** $x \in \mathbb{R}^8$
2: $d \leftarrow \text{floor}(x)$
3: $x_2 \leftarrow d + 0.5$
4: $g \leftarrow (x > x_2)$
5: $s \leftarrow 2 \cdot g - 1$
6: $x_1 \leftarrow d + g$
7: $dist \leftarrow (x - x_2) \cdot s$
8: $\Delta \leftarrow \sum_i dist_i$
9: **if** $\sum_i x_{1,i}$ is odd **then**
10:      $pos = \text{argmin} \, dist$
11:      $x_{1,pos} \leftarrow x_{1,pos} - s_{1,pos}$
12:      $\Delta \leftarrow \Delta + 2 \cdot dist_{pos} - 1$
13: **end if**
14: **if** $\sum_i x_{2,i}$ is odd **then**
15:      $pos = \text{argmax} \, dist$
16:      $x_{2,pos} \leftarrow x_{2,pos} + g_{2,pos}$
17:      $\Delta \leftarrow \Delta + 1 - 2 \cdot dist_{pos}$
18: **end if**
19: **if** $\Delta > 2$ **then**
20:      **return** $x_1$
21: **else**
22:      **return** $x_2$
23: **end if**

---

## D. NestQuantM algorithm

Due to the high complexity of $\text{argmin}$ and $\text{argmax}$ operations in algorithm 5 for the fast hardware implementation, we propose a different, more simple version of NestQuant decoding. Instead of using $\text{argmax}$ and $\text{argmin}$ on lines 10 and 15, we always assign $pos$ to be equal to 1 (thus, indicating that we always "flip" the rounding direction of the first coordinate to fix the parity). Note that this change is only applied in the decoding stage, while during encoding we use the full version of the algorithm, which keeps the granular error the same. Let's denote our modified Gosset oracle as $f : \mathbb{R}^8 \to \mathbb{R}^8$.

**Lemma D.1.** *For a vector $x \in \mathbb{R}^8$ and a vector $v \in E_8$, $f(x + v) = f(x) + v$.*

*Proof.* Recall that $D_8 \subset \mathbb{Z}^8$ contains all the integer vector with even sum of coordinates. The modified Gosset oracle $f$ uses modified $D_8$ oracle $g$, and at input $x$ constructs two candidate points $c_1(x) \in D_8$ and $c_2(x) \in D_8 + \frac{1}{2}$. Then, the algorithm chooses the closest point among these candidates to $x$. Note that $c_1(x) = g(x)$ and $c_2(x) = g\left(x - \frac{1}{2}\right) + \frac{1}{2}$.

We will prove that for $u \in D_8$, $g(x + u) = g(x) + u$ for any $x \in \mathbb{R}^8$. Now, let's show the original lemma. Note that since we are choosing the closest candidate point, if the condition on candidate sets $\{c_1(x + v), c_2(x + v)\} = \{c_1(x), c_2(x)\} + v$ holds, then $f(x + v) = f(x) + v$. Now, consider two cases:

1. $v \in D_8$. Then:

$$c_1(x + v) = g(x + v) = g(x) + v = c_1(x) + v$$
$$c_2(x + v) = g\left(x + v - \frac{1}{2}\right) + \frac{1}{2} = g\left(x - \frac{1}{2}\right) + \frac{1}{2} + v = c_2(x) + v$$

2. $v \in D_8 + \frac{1}{2}$. Then, we say that $v = u - \frac{1}{2} = w + \frac{1}{2}$ for $u, v \in D_8$.

$$c_1(x+v) = g(x+v) = g\left(x+u-\frac{1}{2}\right) = g\left(x-\frac{1}{2}\right) + u = g\left(x-\frac{1}{2}\right) + \frac{1}{2} + v = c_2(x) + v$$

$$c_2(x+v) = g\left(x-\frac{1}{2}+v\right) + \frac{1}{2} = g(x+w) + \frac{1}{2} = g(x) + w + \frac{1}{2} = g(x) + v = c_1(x) + v$$

Thus, in both cases the condition on candidate sets holds, and we get $f(x+v) = f(x) + v$.

Now we show that if $u \in D_8$, $g(x+u) = g(x) + u$. Note that when evaluating $g(x+u)$, we will get the same rounding directions as in $g(x)$, since $u$ is an integer vector. Since $u$ has even sum of coordinates, the parity will also be the same. Then, our decision to flip the rounding of the first coordinate will also match as well. Therefore, the vector between original and rounded coordinates will be the same:

$$g(x) - x = g(x+u) - x - u \Rightarrow g(x+u) = g(x) + u$$

$\square$

Let $v$ be the vector we obtain after rounding to $E_8$, $c$ be the lattice coordinates of $v$, and $G$ be the generating matrix of the lattice. The compressed representation is $c \bmod q$, which corresponds to a point $v' = G(c \bmod q)$. The reconstructed point $\hat{v}$ is defined to be $v' - qf(v'/q)$ by the decoding algorithm. Note that $v - v' \in qE_8$. Let's assume that $f(v/q) = 0$. Then:

$$\hat{v} = v' - qf\left(\frac{v'}{q}\right) = v' - qf\left(\frac{v}{q} + \frac{v'-v}{q}\right) = v' - q\left(f\left(\frac{v}{q}\right) + \frac{v'-v}{q}\right) = v' - q \cdot \frac{v'-v}{q} = v$$

We have used Lemma D.1 in the third equality and our assumption in the fourth equality. Given this fact, we conclude that when using NestQuantM for decoding, the composition of encoding and decoding functions (for a fixed $\beta$) is similar to the original NestQuant, except the shaping region has changed to the set of points $v \in E_8$ such that $f(v/q) = 0$. Since $f$ is close to the original Gosset oracle, we expect this region to still capture Gaussian probability density well. In case of overload errors, we are still able to choose a larger value of beta due to multi-beta strategy.

# E. CUDA Kernel Implementation

The decoding algorithm 2 consists of a basis change $p \leftarrow Gc$ and a subsequent computation of the coset of $\frac{p}{q}$,

$$\frac{p}{q} - Q_\Lambda\left(\frac{p}{q}\right).$$

We leverage the asymmetry between encoding and decoding choosing $G$ which is fast to decode. In particular we use

$$G = \begin{pmatrix} 1 & 0 & 0 & 0 & 0 & 0 & 0 & 0 \\ 1 & 0 & 2 & 0 & 0 & 0 & 0 & 0 \\ 1 & 0 & 0 & 0 & 2 & 0 & 0 & 0 \\ 1 & 0 & 0 & 0 & 0 & 0 & 2 & 0 \\ 1 & 4 & 2 & 2 & 2 & 2 & 2 & 2 \\ 1 & 0 & 0 & 2 & 0 & 0 & 0 & 0 \\ 1 & 0 & 0 & 0 & 0 & 2 & 0 & 0 \\ 1 & 0 & 0 & 0 & 0 & 0 & 0 & 2 \end{pmatrix}$$

$G$ is not a basis for $E_8$ but for $2E_8$. We use $G$ because we want to work in integer-lattice as half-integers cannot be represented with integers. Moreover, we represent 8-vector $\vec{x}$ as two 32 bit integers each representing the two parts of the vector $x_0, x_1, x_2, x_3$ and $x_4, x_5, x_6, x_7$.

For reference, the CUDA implementation of the implementation of $p \leftarrow Gc$.

```
1  __device__ void G_q_fast(
2      const uint32_t enc, uint32_t* out0, uint32_t* out1
3  )
4  {
5      uint32_t even = enc & 0x0F0F0F0F;
6      uint32_t odd = (enc & 0xF0F0F0F0) >> 4;
7
8      uint32_t two_even = even << 1;
9      uint32_t two_odd = odd << 1;
10     uint32_t x0 = even & 0xFF;
11     uint32_t concatenator = (1 << 24) | (1 << 16) | (1 << 8) | 1;
12     uint32_t x0_concat = x0 * concatenator;
13
14     uint32_t temp_even = __vadd4(two_even & 0xFFFFFF00, x0_concat);
15     uint32_t temp_odd = __vadd4(two_odd, x0_concat);
16
17     uint32_t s1 = __dp4a(two_even, (uint32_t)0x01010100, (uint32_t)0);
18     uint32_t s2 = __dp4a(two_odd, (uint32_t)0x01010101, s1);
19
20     *out0 = temp_even;
21     *out1 = temp_odd + s2;
22  }
```

The computation is done by decomposing $G$ into a sum of different matrices.

$$
G = \begin{pmatrix}
1 & 0 & 0 & 0 & 0 & 0 & 0 & 0 \\
1 & 0 & 2 & 0 & 0 & 0 & 0 & 0 \\
1 & 0 & 0 & 0 & 2 & 0 & 0 & 0 \\
1 & 0 & 0 & 0 & 0 & 0 & 2 & 0 \\
1 & 4 & 2 & 2 & 2 & 2 & 2 & 2 \\
1 & 0 & 0 & 2 & 0 & 0 & 0 & 0 \\
1 & 0 & 0 & 0 & 0 & 2 & 0 & 0 \\
1 & 0 & 0 & 0 & 0 & 0 & 0 & 2
\end{pmatrix}
= \begin{pmatrix}
1 & 0 & 0 & 0 & 0 & 0 & 0 & 0 \\
1 & 0 & 0 & 0 & 0 & 0 & 0 & 0 \\
1 & 0 & 0 & 0 & 0 & 0 & 0 & 0 \\
1 & 0 & 0 & 0 & 0 & 0 & 0 & 0 \\
1 & 0 & 0 & 0 & 0 & 0 & 0 & 0 \\
1 & 0 & 0 & 0 & 0 & 0 & 0 & 0 \\
1 & 0 & 0 & 0 & 0 & 0 & 0 & 0 \\
1 & 0 & 0 & 0 & 0 & 0 & 0 & 0
\end{pmatrix}
+ \begin{pmatrix}
0 & 0 & 0 & 0 & 0 & 0 & 0 & 0 \\
0 & 0 & 2 & 0 & 0 & 0 & 0 & 0 \\
0 & 0 & 0 & 0 & 2 & 0 & 0 & 0 \\
0 & 0 & 0 & 0 & 0 & 0 & 2 & 0 \\
0 & 0 & 0 & 0 & 0 & 0 & 0 & 0 \\
0 & 0 & 0 & 0 & 0 & 0 & 0 & 0 \\
0 & 0 & 0 & 0 & 0 & 0 & 0 & 0 \\
0 & 0 & 0 & 0 & 0 & 0 & 0 & 0
\end{pmatrix}
$$

$$
+ \begin{pmatrix}
0 & 0 & 0 & 0 & 0 & 0 & 0 & 0 \\
0 & 0 & 0 & 0 & 0 & 0 & 0 & 0 \\
0 & 0 & 0 & 0 & 0 & 0 & 0 & 0 \\
0 & 0 & 0 & 0 & 0 & 0 & 0 & 0 \\
0 & 2 & 0 & 0 & 0 & 0 & 0 & 0 \\
0 & 0 & 0 & 2 & 0 & 0 & 0 & 0 \\
0 & 0 & 0 & 0 & 0 & 2 & 0 & 0 \\
0 & 0 & 0 & 0 & 0 & 0 & 0 & 2
\end{pmatrix}
+ \begin{pmatrix}
0 & 0 & 0 & 0 & 0 & 0 & 0 & 0 \\
0 & 0 & 0 & 0 & 0 & 0 & 0 & 0 \\
0 & 0 & 0 & 0 & 0 & 0 & 0 & 0 \\
0 & 0 & 0 & 0 & 0 & 0 & 0 & 0 \\
0 & 2 & 2 & 2 & 2 & 2 & 2 & 2 \\
0 & 0 & 0 & 0 & 0 & 0 & 0 & 0 \\
0 & 0 & 0 & 0 & 0 & 0 & 0 & 0 \\
0 & 0 & 0 & 0 & 0 & 0 & 0 & 0
\end{pmatrix}
$$

The next step is to compute $\frac{p}{q}$ and the coset of $\frac{p}{q}$ we merge both operations into one function.

```
1  __device__ void decode_nestquant(
2      uint32_t *enc, int *B_local_decode, const int N = 8
3  )
4  {
5      uint32_t inp0, inp1;
6      unsigned int dist[2];
7
8      const uint32_t MASK_FRACPART = 0x1F1F1F1F;
9      const uint32_t MASK_INTPART = 0xE0E0E0E0;
10     const int HALF_CONCAT = 0x10101010;
11     const int TWO = 0x40;
12     const int ONE = 0x20;
13     const int HALF = 0x10;
14
15     G_q_fast(enc[0], &inp0, &inp1);
16
17     int32_t fracPart0 = inp0 & MASK_FRACPART;
```

```
18      int32_t fracPart1 = inp1 & MASK_FRACPART;

19

20      int32_t integerPart0 = inp0 & MASK_INTPART;
21      int32_t integerPart1 = inp1 & MASK_INTPART;

22

23      int g0 = (fracPart0 & HALF_CONCAT) << 1;
24      int g1 = (fracPart1 & HALF_CONCAT) << 1;

25

26      int32_t change = ((g0 & ONE) - HALF);
27      int32_t sum1 = get_sum(integerPart0, integerPart1);
28      int32_t two_parity_1 = (sum1 & ONE) >> 4;
29      int32_t fracPart0_1 = __vsub4(fracPart0, (int32_t)(two_parity_1 * change) & 0xFF);

30

31      int32_t sum2 = get_sum(g0, g1) + sum1;
32      int32_t two_parity_2 = (sum2 & ONE) >> 4;
33      int32_t fracPart0_2 = __vadd4(fracPart0, (int32_t)(two_parity_2 * change) & 0xFF);

34

35      dist[0] = __vabsdiffs4(fracPart0, HALF_CONCAT);
36      dist[1] = __vabsdiffs4(fracPart1, HALF_CONCAT);
37      int Delta = get_sum(dist[0], dist[1]);

38

39      int32_t dist00 = (dist[0] & 0xFF);
40      Delta -= (two_parity_1 + two_parity_2) * dist00;
41      Delta += two_parity_1 << 4;

42

43      if (Delta <= TWO) {
44          B_local_decode[0] = __vsub4(fracPart0_1, HALF_CONCAT);
45          B_local_decode[1] = __vsub4(fracPart1, HALF_CONCAT);
46      } else {
47          B_local_decode[0] = __vsub4(fracPart0_2, g0);
48          B_local_decode[1] = __vsub4(fracPart1, g1);
49      }
50  }
```

Remember that for $q = 16$, we would need to divide $p$ by $16$. We used a basis for $2E_8$, and not $E_8$ so every element is doubled. This is why in our code $\frac{1}{2}$ is represented by $16$, $1$ is represented by $32$, and $2$ is represented by $64$. In order to avoid using the round function explicitly (which is not parallelized in hardware) we compute the integer and fractional part of the vector using masks (as can be seen in lines 17-21).

We compute $g$ in two parts $g_0$ and $g_1$ (see lines 23 and 24). This is done by checking that the fractional part is greater or equal to $\frac{1}{2}$ (equivalently in our scaled basis, that the fractional part has the fourth LSB set to one). We leverage the fact that

$$x - (\lfloor x \rfloor + 0.5) = \{x\} - 0.5$$

(lines 44 and 45) and

$$x - \text{round}(x) = \lfloor x \rfloor + \{x\} - (\lfloor x \rfloor + g) = \{x\} - g$$

(lines 47 and 48). To compute the bitflip, we compute $2g_1 - 1$ (corresponding to the first element in the vector, see line 26 for reference). We add (subtract) it from fractional part based on the parity check we compute in lines 28 and 32 respectively.

To decode betas in our kernel, we encode the betas as indices to a predefined dictionary (of size 4). Thus, each beta can be represented using 2 bits.

```
1  int beta1 = (beta_packed >> shift) & 0x3;
2  int beta2 = (beta_packed >> (shift + 2)) & 0x3;
3  int decoded_beta1 = beta_dict[beta1];
4  int decoded_beta2 = beta_dict[beta2];
```

Encoding and decoding kernels share the same logical structure so that the complex mapping defined for beta is shared (this allows fast contiguous reads of multiple betas at once).

*Table 4.* Runtime comparison of GEMV kernels on an $8192 \times 8192$ matrix using an NVIDIA A100 GPU.

| Method | Time (μs) |
| --- | --- |
| Baseline (16 bits) | 97 |
| NestQuantM (4.25 bits) | 60 |
| QuIP# (2 bits) | 38 |
| QuIP# (4 bits) | ~75 |
| int4 uniform | 31 |

### E.1. Runtime comparison of GEMV

The QuIP# computation involves invoking two calls to QuIP# (2 bits), so we extrapolate the running time based on QuIP# (2 bits).

## F. Dynamic programming for optimal $\beta$

Recall that instead of using one instance of lattice codebook $C$, we use a union of codebooks $C$ scaled to different coefficients. Specifically, our final codebook $\mathcal{C}$ is parameterized by coefficients $\beta_1 \leq \beta_2 \leq \ldots \leq \beta_k$, and is equal to:

$$\mathcal{C} = \beta_1 C \cup \beta_2 C \cup \ldots \cup \beta_k C$$

Given a set of 8-vectors to quantize, we can find the set of $\beta$ that minimizes reconstruction error using a dynamic programming algorithm, which is described in Appendix F.

When quantizing a vector to the $i$-th scaled codebook, we could either get a small granular error when the vector is in $V_{\beta_i \Lambda}(0)$, or a large overload error otherwise. If we use a codebook with smaller $\beta$, we have larger chance of overload error, but the expected magnitude of granular error is smaller due to the volume of Voronoi region being smaller (Figure 7). We can have two strategies for encoding:

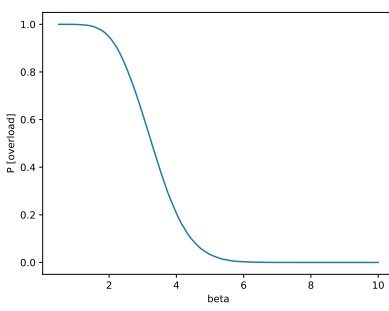

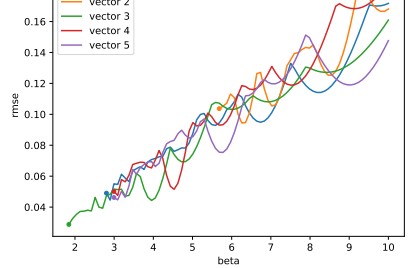

(a) Overload error probablity      (b) Granular RMSE for five sampled vectors

*Figure 7.* Granular and overload error for standard Gaussian vectors, $q = 16$

1. **First-$\beta$:** Use the smallest $\beta$, which does not result in an overflow error.

2. **Opt-$\beta$:** Try all the values of $\beta$, and choose the one that has the smallest reconstruction MSE.

Even though Opt-$\beta$ should provide smaller error, the definition of First-$\beta$ will be useful for us. We can note that the difference between error for Opt-$\beta$ and First-$\beta$ is not very siginificant (Table 5). Moreover, First-$\beta$ can be used to determine the optimal set of $\beta_i$ to use.

Let we have $n$ samples $v_1, v_2, \ldots, v_n$ from the distribution of vectors we are quantizing, and a large set of betas $B$, containing $\beta_1 < \beta_2 < \ldots < \beta_m$, from which we want to take the optimal subset of size $k$ which minimizes the loss under

| K | 2 | 4 | 6 | 8 | 10 |
|---|---|---|---|---|---|
| OPT-$\beta$ | 0.0878 | 0.0795 | 0.0708 | 0.0669 | 0.0646 |
| FIRST-$\beta$ | 0.0878 | 0.0798 | 0.0712 | 0.0676 | 0.0656 |

*Table 5.* Mean RMSE for reconstructed iid standard Gaussian 8-vectors, $q = 16$, $k$ betas are uniform on $[0, 10]$.

First-$\beta$ strategy. For each vector $v_i$ and beta $\beta_j$ we compute $mse_{ij}$ — the MSE if we use $\beta_j$ to quantize $v_i$ and $overload_{ij}$ — whether an overload error occurs in this scenario.

We solve the optimization problem with dynamic programming. Let's define $dp_{ij}$ be the mimimum sum of MSE we can get if we have to quantize all the vectors which do not yield an overload error for $\beta_i$, using $\beta_i$ and $j - 1$ smaller betas and First-$\beta$ strategy. If $i$ is large enough so that no vector has an overflow error on $\beta_i$, $dp_{ik}$ has the answer to the problem. To compute the value of $dp_{ij}$, we can iterate over $s$ — the index of second largest beta in the set (the largest being $\beta_i$). Then, the recalculation works in the following way:

$$dp_{ij} \leftarrow \min\left(dp_{ij}, dp_{s,j-1} + \sum_{p,cond_p} mse_{pi}\right)$$

where $cond_p = overload_{ps} \wedge \neg overload_{pi}$

By following the transtions in this dynamic programming, we can reconstruct the optimal set of $\beta$.

---

**Algorithm 6** Dynamic programming for finding the set of $\beta$

---

1: **Input:** vectors $v_i$, beta set $B$, $mse_{ij}$, $overload_{ij}$
2: $dp_{i,j} = \infty$ for $i$ in $0 \ldots m$, $j$ in $0 \ldots k$
3: $from_{i,j}$ = null for $i$ in $0 \ldots m$, $j$ in $0 \ldots k$
4: $dp_{0,0} = 0$
5: **for** $i = 1$ **to** $m$ **do**
6:     **for** $j = 1$ **to** $k$ **do**
7:         **for** $s = 0$ **to** $i - 1$ **do**
8:             $cond_p = overload_{ps} \wedge \neg overload_{pi}$ for $p \in 1 \ldots n$
9:             $cost = \sum_p cond_p \cdot mse_{pi}$
10:             **if** $dp_{ij} > dp_{s,j-1} + cost$ **then**
11:                 $dp_{ij} \leftarrow dp_{s,j-1} + cost$
12:                 $from_{ij} \leftarrow s$
13:             **end if**
14:         **end for**
15:     **end for**
16: **end for**
17: Let $pos$ is chosen so that $\beta_{pos}$ has no overflow errors
18: result = [ ]
19: **for** $j = k$ **downto** $1$ **do**
20:     result.append($pos$)
21:     $pos \leftarrow from_{pos,j}$
22: **end for**

---

# G. Llama experiment details

We choose the train split of the Wikitext2 (Merity et al., 2017) dataset as a calibration dataset for computing $H$, and evaluate the model on the validation split, computing the perplexity metric. For step 2 in the algorithm (Section 4.6), we

select $\hat{\beta} = [3.5, 4.5, 6.0, 14.5, 25.0]/q$, because it is the $\beta$ we get when optimizing them for weight quantization without consideration of LDLQ. The overall universe of betas contains values from $1$ to $40$ with spacing ranging from $0.25$ to $2$. For running DP on activations, keys, and values, we run the model on a batch of 6 full-length sequences, which is sufficient for this low-dimensional hyperparameter.

When choosing maximum beta for given distribution, we add a margin of $\frac{3.0}{q}$ for weights and $\frac{4.0}{q}$ to the maximum beta needed to have $0$ overload errors on known data to account for potential overload errors in unknown data. While small number of overload error does not affect perplxity significantly, we still aim to minimize their probability.

When computing perplexity for Wikitext2 with given context length, we average the perplexities for all the positions, which is standard for other works in quantization of LLMs.

## H. Ablation studies

We found LDLQ to be useful in improving the quality of quantized model. In table 6, we compare the wikitext2 perplexity of models with and without LDLQ.

| Algorithm | W | W + KV | W + KV + A |
|---|---|---|---|
| NestQuant | 6.308 | 6.379 | 6.633 |
| NestQuant (no LDLQ) | 6.528 | 6.605 | 6.849 |

*Table 6.* Effect of LDLQ on NestQuant ($q = 14$ and $k = 4$) wikitext2 perplexity

While Hadamard matrices from Sylvester construction are commonly used in other works (QuIP#, Quarot), there are multiple ways to construct a fast rotation for the case when dimension is not a power of 2 (such as the down projection in MLP of Llama-3). We tested three possible options for rotation on $q = 14$, $k = 4$, W + KV + A quantization.

| Algorithm | W + KV + A |
|---|---|
| Fourier | 6.773 |
| $S \otimes H$, $S$ — orthogonal, $H$ — Sylvester Hadamard | 6.770 |
| $H_1 \otimes H$, $H_1$ — hardcoded Hadamard, $H$ — Sylvester Hadamard | **6.663** |

*Table 7.* Effect of rotation on NestQuant ($q = 14$ and $k = 4$) wikitext2 perplexity

### H.1. The choice of $k$

The value of $k$, i.e. the number of scaling coefficients is an important hyperparameter of the algorithm. With an increase of $k$, we decrease the quantization error by allowing each vector to be quantized to the lattice point with a proper scaling. However, it increases the bitrate and makes the encoding slower, since we need to try a larger number of scaling coefficients.

We used $k = 3, 4, 5, 8$ to quantize Llama-3-8B across different values of $q$, plotting the resulting perplexity against bitrate in Figure 8. We can see that using $k = 3$ leads to a suboptimal performance of the quantization scheme, while the performances of $k = 4, 5, 8$ are comparable. In our experiments, we use $k = 4$, because lower having $k$ results in faster encoding.

## I. Results for Llama3.2-1B

Here, we show the results of NestQuant on the newer 1B parameter model LLama3.2-1B. We do experiments in the same setups as for the Llama-3-8B model, computing the wikitext2 perplexity.

| q | Bits | Bits (no zstd) | W | W + KV | W + KV + A |
|---|---|---|---|---|---|
| 14 | 3.99 | 4.06 | 10.061 | 10.529 | 11.197 |
| 12 | 3.76 | 3.837 | 10.178 | 10.862 | 11.910 |
| 10 | 3.50 | 3.57 | 10.377 | 11.552 | 14.191 |
| 8 | 3.18 | 3.25 | 10.850 | 13.309 | 18.710 |

*Table 8.* Wikitext2 perplexity of NestQuant quantization of Llama-3.2-1B. The format of the table is the same as in Table 3. The perplexity of non-quantized model is 9.749

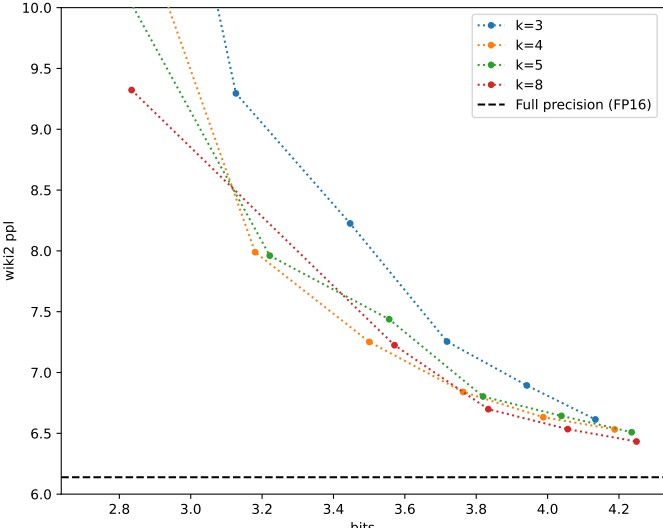

*Figure 8.* The perplexity-bitrate scaling of NestQuant with different values of $k$, all components of the model (weights, KV cache, activations) are quantized

## J. Results for 3-bit model quantization

We publish the results for 3-bit quantization of weights and activations on small models (Llama-3-8B and Llama-2-7B). We use $q = 7$ and $k = 4$, which results in 2.98 bits per entry.

| Bits (W-A-KV) | Method | Llama-2-7B | Llama-3-8B |
|---|---|---|---|
| 16-16-16 | Floating point | 5.47 | 6.14 |
| 4-4-16 | NestQuant | 5.64 | 6.56 |
| 3-3-16 | NestQuant | 8.25 | 6.33 |

