# OpenReview forum: "NestQuant: nested lattice quantization for matrix products and LLMs"
_ICML.cc/2025/Conference — ICML 2025 poster_

### Official Review · Reviewer_c956 · 2025-03-06

**Overall Recommendation:** 3

**Summary:**

This work presents a matrix multiplication replacement for low precision quantized neural networks. A new vector quantization mapping is explored to maximize the efficiency of low bit number distribution. The authors propose to use a new encoding scheme to achieve near lower-bound compression.

**Claims And Evidence:**

The claims made in the submission are supported by extensive existing literature.

**Essential References Not Discussed:**

None

**Experimental Designs Or Analyses:**

See above.

**Methods And Evaluation Criteria:**

The authors evaluated their results on LLaMA3-8B, across several zero-shot reasoning tasks, accuracy, and a perplexity metric on the Wiki2 dataset. The major experiments are provided in Table 1, and more ablation studies are provided in Section 5.

There are many other experiments that need to be performed to demonstrate this method's effectiveness.

1. Results on LLaMA2, including 7B/13B/70B where most existing literature has verified.

2. Results on LLaMA3-70B, where QuaRot, SpinQuant also have provided the results.

In addition to models, the authors need to verify the actual latency on GPUs with dedicated kernels, Section C only provides simulated speedup, which is not enough.

Moreover, for the weights only quantization, 2-bit quantization on all models mentioned above is needed.

**Other Comments Or Suggestions:**

The notation systems could be improved with a Table. Current notation involves multiple symbols yet lacks clear explanation.

**Other Strengths And Weaknesses:**

Weakness

- SpinQuant is referred to as the uniform quantization baseline. However, SpinQuant does not use random Hadamard rotation, I believe the uniform quantization baseline is QuaRot.

- Sec 4.3 overlaps QuaRot, and the authors do not mention their differences.

- I am curious to see if the authors use E8 lattice in QUIP# and align all other setups like random rotation, how much performance improvement can we get?

**Questions For Authors:**

No

**Relation To Broader Scientific Literature:**

This work is related to the model compresion literature.

**Theoretical Claims:**

This work does not introduce new theories.

---

> ### Author Rebuttal · Authors · 2025-04-01
>
> We thank the reviewer for the comment and appreciate the provided feedback. Following the suggestions, we run the evaluations on a larger set of models and provide the perplexity results, as well as comparisons with other methods. The results are outlined in the table in the response to reviewer a2PJ. Our method demonstrates advantage in both weight-only quantization setup, and weight, KV cache, and activation quantization setup. In fact, the models where weights and KV cache were quantized with NestQuant are competitive even with weight-only quantization benchmarks.
>
> For speedup measurements, we introduce a relaxed version of the algorithm, called NestQuantM. In this algorithm, the E8 oracle in dequantization prodcedure fixes the parity of coordinate sum by adjusting the first coordinate, as opposed to picking the best coordinate using argmax/argmin. Such a simplification preserves the correctness of the encoding-decoding procedure (with a slight distortion of the shaping region), but significantly improves the efficiency of the code on GPU. We tested NestQuantM on several models and did not observe significant deterioration in quantization quality, as shown in the ppl results table. We have implemented a CUDA kernel for NestQuantM. When running encode_e8 function, the kernel stores a scaled 8-dimensional vector as a collection of 8 int8 entries, packed into two 32-bit integers, and performs mass operations using regular integer operations and CUDA integer SIMD instructions. Please check the detailed description and benchmarking results in the response to reviewer r651.
>
> We provide the comparisons with QuIP# in the response to reviewer a2Pj. We note that NestQuant outperforms or matches QuIP# performance on all models. In addition, QuIP# uses an inter-layer finetuning algorithm, while NestQuant assumes no access to the data except for Hessian calibration data. The focus of this work is on quantization of W+KV or W+KV+A . These setups are considerably different than W-only quantization (for which QuIP# was designed), because KV and A quantization is *done in runtime* and the encoding algorithm must therefore be fast. For W-only quantization, encoding is done offline, and may be slow.
>
> Having said that, our experiments show that NestQuant attains SOTA ppl even for W-only quantization at R=4 bits. Going to the extreme low-rate regime, e.g., R=2 bits,  requires further ideas from O&P24 not discussed in the present manuscript (MMSE scaling + sparsification + dithering). We will implement them, but it takes more time, and the experiments may not converge in the rebuttal time-frame.
>
> We also respond to the remaining comments:
>
> - We treat both QuaRot and SpinQuant (as well as other quantization schemes) as our benchmarks, and show improvement with respect to all previous quantization schemes. In Figure 3, we show results for synthetic data consisting of Gaussian iid matrices. Since the Gaussian iid distribution is isotropic, any rotation applied on it will not change its distribution, and SpinQuant and QuaRot are identical in this context, because (while they apply different rotations) they both use uniform quantizers.
>
> - We did mention in Section 2.2 that QuaRot already applied Hadamard transform, but the reviewer is right and we should have also mentioned this in Sec 4.3. We apologize for the omission. In a revision Sec. 4.3 will be sure to reference prior literature on rotations (SliceGPT, QuIP, QuaRot and SpinQuant)”
>
> - As mentioned above, QuIP# was designed as a W-only quantization method. Consequently, its dequantization algorithm was designed to run fast, but not its quantization algorithm. Our experiments show (small) improvement over QuIP# even in the W-only quantization regime, which is not the focus of our work. Conducting experiments for W+KV/W+KV+A quantization  with our Alg 1 + Alg 2 + overload avoidance mechanism replaced with the QuIP# E8P quantization/dequantization from Sections 4.2 & 4.3 in QuIP# paper will take a long time, because the QuIP# E8P quantization algorithm is not fast enough. Given the fact that NestQuant attains better ppl for W-only quantization, we expect the same trend also for W+KV and W+KV+A.

---

> > ### Comment · Reviewer_c956 · 2025-04-05
> >
> > I'd like to thank the authors for their rebuttal, I have checked the authors' reply and review comments from other reviewers. I am also very familiar with SOTA and understand LLMs quantization topic. First of all, I do not appreciate how this paper is drafted, mainly because of its chaotic notation system. The matrix/sets are both represented by capital letters, sometimes, a set is represented with calligraphic letters. Moreover, many notations used in former texts are not used later. I understand that everyone has their preference of a manuscript and some may like this style (and I respect that). But personally speaking I do not like how the problem and solution are presented in this paper.
> >
> > Additionally, I want to ask a few more questions regarding the rebuttal before I make the final decision.
> >
> > Q1. Throughout this paper, including sec 2.1, figure 2, the authors talk about how to quantize weights and activations simultaneously using NestQuant. However, the experiments do not show W4A4 (or W3A3) results. Why spend so much space for W and W+KV experiments?
> >
> > Q2. Following Q1, I am confused about how NestQuant can quantize activations with practical benefits on hardware. It seems like activations require decoding before the matmul operations, which cannot remain in floating-point precision (correct me if I am wrong). Then, in this case, the major motivation is to find a fast decoding and accurate algorithm.
> >
> > Q3. Regarding the cuda kernel, the authors use GEMV, which is for decoding. However, based on the QuaRot paper, " *As the prefill stage is known to be compute-bound [Ashkboos et al., 2023], joint quantization aims to reduce the precision of parameters and KV cache (which results in lower memory usage) as well as inputs (known as activations) and compute the forward pass in low precision* ", the purpose of weight activation quantization benefit should be on prefill stage. It seems NestQuant cannot do that.
> >
> > Q4. Following Q3, the authors mentioned, " *Our CUDA Kernel does GEMV (matrix-vector) multiplication, assuming the matrix is quantized with NestQuantM and the vector is unquantized* ", Does that mean quantizing the activation (vector) is not necessary since you need to decode it into floating-point in decoding?
> >
> > By the way, regarding the concerns on comparing QuIP#, thank you for your clarification. I see now that the encoding speed is the main difference. However, I think the motivation of this paper is misleading. The current motivation makes the reader think you can accelerate matmul rather than encoding the dynamic activation vectors.
> >
> > Finally, thanks again for providing results on other models, they are helpful to understand the performance of NestQuant.

---

> > > ### Author Response · Authors · 2025-04-08
> > >
> > > We appreciate the reviewer’s comment on our response. We believe that it would be worthwhile to re-emphasise the motivations and the practical implications of NestQuant. These motivations are outlined in the beginning of the Introduction section of the original paper.
> > >
> > > Our view is that the primary aim of quantization is to reduce the memory consumption enabling usage of cheaper GPUs for inference (less VRAM), improve storage efficiency of cold sessions (KV cache) and allowing bigger batches with QLORA (quantized forward pass activations) or pipelined parallelism in distributed inference. The secondary aim is accelerating inference by saving HBM load times in the generation phase. Since during the latter the matrix-vector is the dominant operation, we implemented the GEMV kernel (see response to reviewer r651) to demonstrate that our dequant function has reasonable latency as well (i.e. 4.25-bit NestQuant is slower than 2-bit QuIP# but faster than 4-bit QuIP#).
> > >
> > > We want to stress that our work clearly shows that NestQuant yields better perplexity for a given budget of RAM, while inference speed will depend on the particular HBM speed and other hardware details. We do hope, though, that our demonstrations of strong perplexity gains will encourage work on hardware-assisted quant/dequant in future TPUs and GPUs.
> > >
> > > We also wanted to mention that as you requested, we obtained the Llama-3-70B quantization result at W4A4KV4 ppl=3.70 (next best of all is OstQuant at 4.01).
> > >
> > > Q1. The importance of W-only and W+KV experiments is mainly on saving VRAM. However, we are also accelerating the inference – see the table we posted before, our kernel runs at int8-speed while taking int4-memory and SOTA perplexity. We didn’t present W4A4 results because W4A4KV4 already outperforms SOTA on W4A4 for most models. However, we have ran experiments and show W4A4 results and preliminary results on W3A3.
> > >
> > > |       | Llama-3-8B | Llama-2-7B |
> > > |-------|------------|------------|
> > > | W4A4  |    6.56    |       5.64     |
> > > | W3A3  |     8.25       |     6.33       |
> > >
> > > Q2. Quantizing activations is useful for QLORA, for future guidance of hardware-assisted low-precision storage (energy efficiency) and for pipelined parallelism. We reiterate that our work demonstrates clear advantages (in perplexity) for using NQ for both Weights and W+KV+A quantization. Note that other vector quantization algorithms (AQLM, QuIP#, QTIP) have quantization functions that are too slow for runtime implementation, thus those algorithms are only used as W-only methods (though we are competitive even in this mode).
> > >
> > > Q3. Using GEMV, we demonstrate that loading (smaller in size) quantized matrix and performing dequantization is faster than loading unquantized matrix. This should offer advantage of W-only quantization in both prefill and generation, although we expect the relative speedup in generation to be larger. We also note that due to properties of E8, dequantization results in scaled int8 vectors, which allows us to use int8 multipliers for matmul. This can be seen by the reviewer in our GEMV code, which uses __dp4a() CUDA calls (!).
> > > BTW, there was a slight mistake in our CUDA kernel, updated version runs slightly slower at 58us, code: https://pastebin.com/7G9EXgTc
> > >
> > > Q4. The CUDA kernel is for the simplest version of NestQuant quantized matrices (W or KV, for example) and plain int8 activations (e.g. queries). This would correspond to W4KV4A8 case (which we did not test). For the reported results of W4KV4A4, the activations themselves are stored via NestQuant in 4 bits/entry. In this case the CUDA kernel requires another call to decode_nestquant() to unpack NQ-stored activations into int8-activations.
> > >
> > > We apologize for being not clear enough in response to reviewer r651, “unquantized” here means that we didn’t use NestQuant for quantization of activation vector, but we still store it in 8 bits with uniform quantization. We use same setup when assessing QuIP#’s speed.

---

### Official Review · Reviewer_c7kq · 2025-03-10

**Overall Recommendation:** 5

**Summary:**

The paper contributes a novel practical vector quantization method and applies it to quantize LLMs (post training). It is based on information theory with the key elements being:
- Hadamard transforms to bring the distribution of vectors (weights/activations) closer to normality
- Vector quantization of groups of 8 elements using the Gosset lattice. More precisely, a multi-scale Gasset lattice, where the set of scales is optimally estimated on a small calibration sample.
The method has been shown to achieve better LLM performance when quantizing weights and keys / values to 4 bits, compared to current SOTA, and the analysis suggests its practicality allowing also a speed-up besides memory savings.

## update after rebuttal

I had several concerns on the clairity, technical details and the how the reasoning is specific to preserving matrix products. The concerns have been resolved by the rebuttal, they can be addressed with a minor revision. From the experimental side, from my point of view, the paper presents enough evidence for improving SOTA quantization quality of large models and reasonable evidence of potentially low computation costs. It should certainly be accepted.

**Claims And Evidence:**

I have not noticed any over-claiming or inconsistency in the claims and the experiments.

**Essential References Not Discussed:**

**Q3** I think I would like to see more clearly what constructs in the paper are claimed innovative / original. In particular, a more detailed discussion of what is different in the methods from Tseng et al. 2024. They seem to be using Gosset lattice E8 and Hadamard transforms as well. Why exactly their method is deemed impractical by the paper?
Also, what is the attribution of the encoding-decoding algorithms 1,2 for the lattice restricted to a cell of $q \Lambda$?

**Experimental Designs Or Analyses:**

I find the experiments to be sound, comparing the perplexity at the same bits per entry for all methods in Table 1 and in Fig 1 for several rates. Llama results for different rates in Tables  2,3 should provide a broader basis for potential applications and further methods. Validation in Fig 5 takes into account both the perplexity and the change in the code length due to different k. Additional experiments on a synthetic problem (Fig 3),  and ablations in Tables 5,6 are also very useful. Perhaps it would be instructive to see what happens if orthogonal transforms are omitted.

**Methods And Evaluation Criteria:**

Yes

**Other Comments Or Suggestions:**

# Minor Issues

035R: Would be useful to elaborate on what is KV cache, and why it is a bottleneck, regarding the memory.

085R: It is not clear what is random for the expectation.

173: How finding the nearest lattice point is an assumption?

220: I find it hard to follow. Choosing $S \in R^n$ and independently choosing $\beta$ to scale $S$ seems redundant.

230: What is the difference between ${\rm covol}(\Lambda)$ and ${\rm vol}(\mathcal{V}_\Lambda)$ in (3)?

236: $vol(B)$?

245: One confusion I have is that lattice quantization is still uniform over each of its basis directions. The claim was rather obscure. It would help to detail the example in Fig 2: show the Generator, is this lattice a subset of integer grid if scaled? I guess not?

I've noticed Gosset lattice has symmetries with respect to 4x4 Hadamard transform. Then perhaps the rotation can be simplified a bit building from $I_4$ blocks? I am not sure this makes sense, but though I would mention it anyway.

**Other Strengths And Weaknesses:**

**Q4** In Fig. 4. Why there is no quantization of queries (top path after orange Hadamard). Is it intentional, like in this is in-chip and not a computation bottleneck? Then I wonder why quantizing the values (bottom path) is without a Hadamard before it? Doesn't it degrade the quantization accuracy? In fact, from this diagram, I don't know why the whole bottom path till the pooling with attention weights cannot be just identity, because I think a linear transform, which is the same for all tokens, can be as well applied after the pooling.
The notation of the plot is confusing. I assume a group like $H W_k$ means $H$ applied to weights and then quantized (because of the Q supperscript). Also, there are many different colors in the Figure. I they have a purpose, it would be appropriate to explain in the caption. For instance, it would be nice if paired (same) Hadamard transforms were of the same color, but this seems not to be the case for the first on the left and those applied to $W_q$, $W_k$, $W_v$.

**Q5** Can quantizing activations with a Hadamard transform actually achieve a memory saving, i.e. can the Hadamard transform be applied on the fly, without writing the activations to memory before quantizing them? If not, then there is a write and read of them involved, and they need to be decoded before used in the linear layer. It seems to me there is no advantage in memory use or speed compared to leaving activations as int8.

**Questions For Authors:**

I marked the most important questions as Q1-5 above. In summary, I am concerned with that not everything is presented clearly enough and possibly somewhat misleading. I am open to rising the score if these questions are clarified / rebutted.

**Relation To Broader Scientific Literature:**

I am no expert, but what I see in the paper, it mentions many works in various contexts (e.g. locality-sensitive hashing).  The paper advocates quantization for approximating inner products and cites recent theoretical results in 124. As discussed above, it is not entirely clear whether this is actually used towards the design of the scheme. I do not see the paper to be contributing substantially in methods, so I would not expect it to impact in the broader scope.

**Theoretical Claims:**

The paper devotes significant space to review prior work and background. It is not quite clear to me which parts of section 3 present original ideas. Mainly it is probably adaptation of results from information theory. I have some concerns about clarity, but they are also connected with the theoretical claims made.

First, I find the paper lacks clarity in defining the following:
1) Lattice. It is specified rather late in 308 and indirectly.
2) nested lattice (139, 176) / nested lattice quantization. Should be important, given the title, right? Is nested lattice a lattice?
3) Gosset lattice. It is specified in 324, but I do not see how is it a subset of standard integer lattice as claimed in 265-267. It is also not obvious that $D  \cup D + \frac{1}{2}$ is a lattice.

110: I am confused by the claim that $UX$ is i.i.d. Gaussian, i.e. $UX \sim N(0,I)$, because this necessarily implies that $X\sim N(0, I)$ as well.

Eq. (1):  I believe the authors mean the hats on X and Y separately.

**Q1** What part of the method makes it specific to approximating the product of matrices rather that optimizing the expected distortion of X and Y separately? It seems to me that all the arguments and constructions in Sections 3/4 (in particular, granular and overload errors) are motivated by the Normality of inputs and the expected distortion. The method for finding optimal scales also seems to be optimizing the expected distortion.

 Related to the above, How does Eq. (1-2) compare with a rate that one gets for $E(X' Y - \hat X' \hat Y)^2$ when quantizing $X$ and $Y$ optimally to minimize the distortions $E (X - \hat X)^2$, $E (Y - \hat Y)^2$? Doesn't it give $2*2^{-2 R}$, which is kind of very similar? I assume we are in the range $R > 1$...

316: It is not clear whether $x$ is assumed in $\Lambda$ or in $C$ at this point. The claim that $Q$ is a bijection between C and $\mathbb{Z}_q^d$ for any generator perhaps needs a justification or reference.

**Q2** I do not see how the algorithm implements the shaping according to the Voronoi region of $q\Lambda$. The Encode procedure in Alg. 1 already assumes $x$ in the Voronoi cell of $q \Lambda$ but in Alg. 3 this is not ensured. The encoder using $v {\rm mod} q$  can of course encode any lattice point (then the input constraint of Alg.1 needs to be lifted). The question is whether the decoder will decode it with zero error iff $x \in V_{q \Lambda}(0)$? I guess yes, but would be nice to clarify in the paper.

---

> ### Author Rebuttal · Authors · 2025-04-01
>
> Thank you for such detailed reading and thoughtful comments! Please refer to other responses for new experiments.
>
> Lattice defs clarity: We will provide references and a short definition of lattices and nested lattices early
>
> 110: The exact statement is only given later in Sec 3-Random Rotation.
>
> Eq. 1: $\widehat{X^T Y}$ is our generic notation for the inner-product $X^TY$ from the quantization bits.
>
> ### Q1
>
> The fact that we can design quantizers for Gaussian matrices, and those work universally for any two matrices $X$, $Y$ is a special feature of the MatMul problem. The reason is that we can apply a rotation $U$ on both $X$ and $Y$, and quantize $UX$ and $UY$ (instead of $X$ and $Y$), without affecting the matrix product. The rotation makes $UX$ and $UY$ essentially Gaussian, so we only need to design quantizers for this distribution.
>
> Not any good vector quantizer (VQ) will work well for MatMul. In VQ only the trace of the quantization error covariance matrix matters. In MatMul quantization, its Frobenius norm is also important (O&P24). For “good” lattice quantizers in any dimension, this Frobenius norm is known to be small, which motivates the use of nested lattice quantizers.
>
> O&P24 show that for $R>0.906$ optimal nested *lattice* quantization of each column vector is indeed optimal for MatMul quantization. For smaller rates, further dimensionality reduction is needed. At this point we focused on just implementing the fast VQ, in the future other ideas from O&P24 will be added. For example, at R=1.2 bit/entry we found that in E8 NestQuant, dropping 10% of vector entries and at R=1 bit dropping 25% of vector entries works best.
>
> We also note that off-the-shelf lattice VQs are either highly suboptimal or too slow for our purposes, and our novel multi-scale trick solves this.
>
> 316: x is only assumed to be in $\Lambda$, and this will indeed be made clear. We will add a reference to C&S’83 for bijection. See also our answer to Q2.
>
> ### Q2
>
> It's true that from $v$ mod $q$ the decoder can only know that the described point belongs to a particular coset of “coarse” lattice $q\Lambda$. However, only one point (our decoder output!) in this coset belongs to the Voronoi cell of $q\Lambda$, thus resolving ambiguity. This is described in C&S’83 and we will add a reference.
>
> ### Q3
>
> As explained in Q2, Alg1 describes a coset of the fine lattice, and Alg2 chooses the unique member in the coset that belongs to the cell of $q\Lambda$. This simple enc/dec procedure is possible only due to the quotient group structure $\Lambda/q\Lambda$ of NestQuant. Shaping regions that are not cells of $q\Lambda$ do not lend themselves to these enc/dec algorithms.
> In our experiments we used $\Lambda=E_8$, as in QuIP#. Note however, that NestQuant enables efficient encoding/decoding of the $2^{dR}$ points from E8, and allows to use any R that is log2(int), which we see as a major advantage. QuIP# uses a different set of $2^{dR}$ points and encodes them via slow residual quantization and LUT. For 2,3,4-bits and W-only quantization (which is the goal in QuIP# and QTIP) this is totally fine, but for KV+A that are quantized in runtime, those solutions are too slow.
>
> The NestQuant framework is not restricted to E8. It can be implemented with any base lattice $\Lambda$. In O&P24 it is shown that for “good” high-dim lattices NestQuant is information-theoretically optimal for MatMul quantization. Practical considerations prohibit high-dim lattices, and instead we use low-dim lattices. In such dimensions, overload errors are unavoidable and deteriorate performance. Our novel overload avoidance mechanism (multi-scale) is crucial for attaining SOTA ppl.
>
> ### Q4
>
> We don’t quantize queries for a fair comparison with other works (SpinQuant, QuaRot), which also keep queries in 16 bits when KV cache is quantized. In autoregressive generation the queries are used once, while keys and values are used by all subsequent tokens. We agree that applying head-wise Hadamard transformation on values would make them easier to quantize and are thankful for this suggestion.
>
> In the final revision, we will clarify Figure 4. The Q-box around $HW_K$ group indeed means that H is applied to the weight and this matrix product is quantized: The $W_oH$ part of the diagram should be corrected to $HW_o$. In this diagram, H matrices on key-query path are applied per-head (i.e. in small dimension), while all the other H are full embedding dimension sized.
>
> ### Q5
>
> We agree that quantizing activations below int8 should be used for the scenario when they need to be sent over the network for very large models under pipelined setup. The answer to the question of whether Hadamard transform can be done on the fly depends on particular hardware and cache structure but we do mention that SpinQuant and others are already using this idea in practice. We believe that one activation vector should fit in the L1 cache of most GPU SMs, hence making fast Hadamard transform especially fast.

---

> > ### Comment · Reviewer_c7kq · 2025-04-03
> >
> > I thank the authors for a detailed response, I think all clarity concerns have been addressed. I have checked also other reviews. I am very fully familiar with SOTA, but to me the experimental verification presented appears quite thorough and convincing. Also, I do not expect ICML papers to deliver optimized CUDA kernels — these aspects affect the practical impact of the work but not as much the research contribution. I have raised my score to strong accept.

---

### Official Review · Reviewer_r651 · 2025-03-12

**Overall Recommendation:** 2

**Summary:**

The authors propose NestQuant, a novel lattice-based quantization method designed to improve weight-activation quantization for LLM.

**Claims And Evidence:**

The authors provide evidence, such as RMSE error comparisons with SpinQuant, to demonstrate the effectiveness of NestQuant relative to uniform-based methods.

**Essential References Not Discussed:**

The paper discusses weight-activation quantization and utilizes rotation transformation, but does not compare or discuss the following related rotation-based PTQ methods:

- DuQuant (NeurIPS 2024)
- OstQuant (ICLR 2025)

Considering their relevance, I recommend including a comparison with DuQuant, and OstQuant can be an additional option for comparison.

**Experimental Designs Or Analyses:**

The primary experiments are conducted on Llama3-8B, with some extension to Llama3.2-1B. However, I am curious about the effectiveness of NestQuant on larger models, such as Llama3-70B, and other series like Qwen2.5-7B.

Additionally, the current experiments do not include speedup measurements, which are crucial, especially considering the use of the new lattice-based quantization. A detailed discussion of these measurements is recommended.

**Methods And Evaluation Criteria:**

The evaluation benchmarks align well with those typically used for PTQ methods.

**Other Comments Or Suggestions:**

More emphasis could be placed on evaluating the method's performance with larger models and its impact on inference speed and memory usage.

**Other Strengths And Weaknesses:**

The lattice-based quantization approach is a promising and interesting concept.

**Questions For Authors:**

1. Could you include a comparison with DuQuant and OstQuant in the paper?
2. Please consider adding more experiments on larger models, particularly Llama3-70B and Qwen2.5-7B.
3. Could you provide a detailed analysis of the inference speedup and memory usage, especially given the new lattice-based quantization method?

**Relation To Broader Scientific Literature:**

There is a lack of comparison with other rotation-based PTQ methods, particularly the following:

1. DuQuant: Distributing outliers via dual transformation makes stronger quantized LLMs, NeurIPS 2024.
2. OstQuant: Refining Large Language Model Quantization with Orthogonal and Scaling Transformations for Better Distribution Fitting, ICLR 2025.

Given the published date, I believe a comparison with DuQuant is essential, while the inclusion of OstQuant would also be valuable.

**Theoretical Claims:**

The theoretical section is clear.

---

> ### Author Rebuttal · Authors · 2025-04-01
>
> Thank you for reviewing our paper and providing valuable feedback! Here, we provide the answers to the questions.
>
> ### **Evaluation on other models**
>
> We have conducted additional experiments of NestQuant in compressing other LLMs. We chose models in Llama-2 series (7B, 13B, and 70B) and Llama3-70B, as most popular in the prior quantization literature (thus facilitating comparison). We considered three regimes: weight only, W+KV, and W+KV+A. We quantize the models to < 4 bits by choosing ($q=14$) for a fair comparison with other methods.
>
> Please refer to the table in response to reviewer a2PJ for Wikitext2 perplexity results with context length 2048. NestQuant consistently achieves better perplexity metrics across different models for both weight-only regime and full quantization. In fact, for all models we have tested, except Llama-2-7B, NestQuant with W4A4KV4 quantization even outperforms previous works with W4A4KV16.
>
> We hope that the new evaluation results would provide more evidence for robustness of the performance advantage of NestQuant.
>
> ### **Comparison with DuQuant and OstQuant**
>
> Indeed we should have included these important references, and we thank the reviewer for this comment. We now compare our numerical results to the results reported in those references under the same setup. In all setups we’ve tested, NestQuant improves over the ppl reported in those papers. On a deeper level, both DuQuant and OstQuant can be viewed as “pre-processing” techniques that create weights and activations that are “easier” to quantize, while the quantization method used there is essentially round-to-nearest (RTN). The main innovation in NestQuant is the ability to use quantizers that are more effective than RTN, at the expense of additional cycles in encoding/decoding complexity. These quantizers are based on “good” nested lattices and on the overload avoidance mechanism. Thus, the approaches in DuQuant/OstQuant and that of NestQuant actually complement each other. Combining the two approaches is an excellent direction for future research that is expected to further improve the SOTA.
>
> ### **Latency measurements**
>
> We acknowledge that the significant improvement that NestQuant achieves in perplexity comes at a cost of slower quantization/dequantization speed. We provided FLOP count in our appendix, which show that the computational burden of NestQuant is reasonable. Creating a fully-optimized CUDA kernel for NestQuant is indeed an important task for future work.
>
> As a temporary demonstration that NestQuant dequantization can indeed be fast on GPUs, we have implemented a CUDA kernel for a suboptimal version of NestQuant, which we refer to as NestQuantM. The difference between NestQuant and NestQuantM is that in the latter, the E8 optimal nearest neighbor decoder is replaced by a sub-optimal decoder. NestQuantM is close in ppl performance to NestQuant and still outperforms SOTA (see the ppl results table). Our CUDA Kernel does GEMV (matrix-vector) multiplication, assuming the matrix is quantized with NestQuantM and the vector is unquantized.
>
> Below, we provide the runtime of NestQuantM GEMV kernel, compared to other baselines. The measurements are made for a $8192 \times 8192$ matrix on A100 GPU.
>
> | Method | Time (us) |
> | -------- | ---------- |
> | Baseline (16 bits) | 97 |
> | NestQuantM (4.25 bits) | 50 |
> | QuIP# (2 bits) | 38 |
> | QuIP# (4 bits) | ~75 |
> | int4 uniform | 31 |
>
> Note that we estimate the runtime of QuIP# (4 bits) as double of the runtime of QuIP# (2 bits), since the size of the matrix that is loaded from DRAM doubles and the E8P dequantization procedure needs to be ran twice due to RVQ. Also, while we use 4.25 bit version with q=16 for benchmarking, the ppl for NestQuantM is still computed with q=14.
>
> This kernel makes NestQuantM a practical quantization scheme for weight and KV cache quantization. We note that uniform quantization algorithms achieve a much higher perplexity values. Other vector quantization methods, such as QuIP#, QTIP, or AQLM, are only usable for weight quantization, since they require a significantly more complex encoding (e.g. QTIP requires solving dynamic programming problem for finding optimal trellis path). Additionally, we note that NestQuant can operate at any rate $R=\log_2(q)$ for integer $q$ with essentially no change in the code.
>
> We also wanted to stress again that the goal of this work was to improve the frontier of the tradeoff between the quality of the model and the amount of bits transferred into/from the compute cores. At this stage, we did not yet focus on improving actual generation speed. In terms of utility, however, we want to mention that transferring one byte from DRAM (HBM) requires 3000x the amount of energy of int8 multiplication (Horowitz, doi: 10.1109/ISSCC.2014.6757323). Thus, even accounting for the extra algorithmic overhead we believe that NestQuant already gives improved energy consumption for a given perplexity compared to other schemes.

---

### Official Review · Reviewer_a2PJ · 2025-03-16

**Overall Recommendation:** 4

**Summary:**

The presented work proposes to use in an improved quantization technique for LLMs that wastes 17% less quantization "space". Written well and very balanced (theory + practical verification).

## update after rebuttal

Initially my own (internal) score was between 3 and 4 and I rounded up to 4, because I found the presentation compelling enough and the results good. The very healthy discussion here reaffirmed me that the submission should be accepted (for me it is now a solid 4) and hope the authors add all the additions discussed to the final version. Thank you to the authors for also actively supporting the rebuttal!

**Claims And Evidence:**

The claim of improvements by using an improved technique is proven by experimental results (up to 20% relative).<

**Essential References Not Discussed:**

Everything good.

**Experimental Designs Or Analyses:**

Good.

**Methods And Evaluation Criteria:**

All standard evaluation criteria have been applied, also perplexity which is known to be most sensitive.

**Other Comments Or Suggestions:**

None.

**Other Strengths And Weaknesses:**

None.

**Questions For Authors:**

None.

**Relation To Broader Scientific Literature:**

As the technique has been inspired by other work (Voronoi regions), I expect influence of a broader audience than just ML.

**Theoretical Claims:**

Validated to be correct. I certainly appreciate the appendices much!

---

> ### Author Rebuttal · Authors · 2025-04-01
>
> We thank the reviewer for their feedback. Indeed we believe that this work may be interesting also to researchers outside the ML community. In particular, NestQuant consists of a combination of various ideas, including random rotations, nested lattice quantization for the induced Gaussian-like vectors, the overload avoidance mechanism with multiple beta values together with Algorithm 6 designed for choosing those values.
>
> These ideas span information theory, communications, and even pure math (the lattices used for MatMul quantization should have properties that were not previously studied - in particular, their covariance matrix should have a small Frobenius norm). We certainly hope this work will initiate a broader interest in PTQ for LLMs within the mentioned research communities.
>
> In addition to the evaluations provided in the paper, we also perform additional evaluations of NestQuant on models from Llama 2 series (7B, 13B, 70B), and Llama-3-70B (though we didn't have enough time to finish simulation of W+KV+A quantization for this model). The table below shows the Wikitext2 (seq. length 2048) perplexities of NestQuant and other PTQ methods, with NestQuant outperforming state of the art for most models and setups.
>
> | Bits (W-A-KV) | Method         | Llama-2-7B | Llama-2-13B | Llama-2-70B | Llama-3-8B | Llama-3-70B |
> |---------------|----------------|------------|-------------|-------------|-----------|-------------|
> | 16-16-16     | Floating point | 5.47       | 4.88        | 3.32        | 6.14      | 2.86        |
> | 4-16-16      | QuaRot         | 5.60       | 5.00        | 3.41        | -         | -           |
> |               | QuIP#          | 5.56       | 4.95        | **3.38**        | -         | -           |
> |               | OstQuant       | 5.64       | 4.94        | 3.41        | 6.53      | 3.19        |
> |               | NestQuant      | **5.53**       | **4.93**        | **3.38**        | **6.31**      | **3.14**        |
> |               | NestQuantM     | 5.55       | 4.95        | -           | 6.35      | -           |
> | 4-16-4              | NestQuant      | 5.57       | 4.96        | 3.39        | 6.37      | 3.19        |
> |        | NestQuantM     | 5.59       | 4.99        | -           | 6.49      | -           |
> | 4-4-16       | SpinQuant      | 5.9        | 5.2         | 3.8         | 7.1       | -           |
> |               | OstQuant       | 5.60       | 5.14        | 3.57        | 7.24      | 3.97        |
> |               | DuQuant        | 6.08       | 5.33        | 3.76        | -         | -           |
> | 4-4-4        | QuaRot         | 6.10       | 5.40        | 3.79        | 8.16      | 6.66        |
> |               | SpinQuant      | 5.9        | 5.3         | 3.8         | 7.3       | -           |
> |               | OstQuant       | 5.91       | 5.25        | 3.59        | 7.29      | 4.01        |
> |               | NestQuant      | **5.67**       | **5.03**        | **3.49**        | **6.63**      | TBD         |
> |               | NestQuantM     | 5.73       | 5.07        | -           | 6.82      | -           |
>
> In this table, there is a method "NestQuantM", which corresponds to NestQuant with a simpler, but more computationally efficient E8 lattice oracle used in the decoding. We evaluated NestQuantM for Llama-2-7B, Llama-2-13B, and Llama-3-8B
>
> We have attached a figure: https://imgur.com/a/bNW1D7K for reference.

---

### Decision · Program_Chairs · 2025-05-01

**Decision:**

Accept (poster)

**Comment:**

The paper presents a novel and effective post-training quantization method that achieves state-of-the-art results for low-precision quantization of large language models. The approach, based on nested lattices, demonstrates significant improvements in perplexity compared to previous methods, including SpinQuant and QuaRot.

Reviewers provided valuable and detailed feedback, especially during the rebuttal phase. While some concerns were well-addressed, some areas still require attention in the final camera-ready version:

- Comparison with DuQuant: The reviewers noted the importance of including comparisons against DuQuant, which the authors partially addressed. It is essential that these comparisons are clearly incorporated into the final version to provide a comprehensive view of the performance landscape.

- Latency verification with dedicated CUDA kernel: Reviewers emphasized verifying actual latency on GPUs using a dedicated CUDA kernel. This verification is crucial to ensure the practical applicability and demonstrate clear benefits in real-world inference scenarios. The authors provide an evidence via NestQuantM kernel and should be expanded and incorporated in the main text.

- Notation and clarity improvements: The methods section currently suffers from notation inconsistencies, with matrices and sets represented both by capital and calligraphic letters interchangeably. Additionally, several symbols introduced earlier in the manuscript are not consistently used in subsequent sections. This must be resolved for better readability and precision.

I strongly encourage the authors to incorporate the above points thoroughly in their final manuscript to enhance clarity, reproducibility, and impact. Overall, the submission represents a significant contribution to the field of LLM quantization and should be accepted.